# Hugin-AstA circuitry is a novel central energy sensor that directly regulates sweet sensation in *Drosophila* and mouse

Wusa Qin[1†], Tingting Song[2,3†], Zeliang Lai[4], Daihan Li[3], Liming Wang[1,2,3]*, Rui Huang[3,5]*

[1]Institute of Molecular Physiology, Shenzhen Bay Laboratory, Shenzhen, China; [2]Laboratory for Clinical Medicine, School of Public Health, Capital Medical University, Beijing, China; [3]Chinese Institutes for Medical Research, Beijing, China; [4]Center for Neurointelligence, School of Medicine, Chongqing University, Chongqing, China; [5]Guangyang Bay Laboratory, Chongqing Institute for Brain and Intelligence, Chongqing, China

## eLife Assessment

This **important** work delineates layered glucose-responsive neuropeptidergic mechanisms that regulate sugar intake. Using a combination of genetic, physiological, and behavioral experiments, the authors **convincingly** show that Hugin- and Allatostatin A-releasing neurons suppress sugar feeding by reducing the sensitivity of Gr5a-expressing gustatory neurons. They further demonstrate that Neuromedin U neurons share key physiological properties with fly Hugin neurons, highlighting conserved peptide functions across animal phyla.

**\*For correspondence:**
lmwang83@cimrbj.ac.cn (LW);
huangrui0716@sina.com (RH)

†These authors contributed equally to this work

**Abstract** Taste sensation plays a crucial role in shaping feeding behavior and is intricately influenced by internal states like hunger or satiety. Despite the identification of numerous neural substrates regulating feeding behavior, the central neural substrate that linked energy-sensing and taste sensation remained elusive. Here, we identified a novel neural circuitry that could directly sense internal energy state and modulate sweet sensation in the *Drosophila* brain. Specifically, a subset of neuropeptidergic neurons expressing hugin directly detected elevated levels of circulating glucose via glucose transporter Glut1 and ATP-sensitive potassium channels. Upon activation, these neurons released hugin peptide and activated downstream Allatostatin A (AstA)[+] neurons via its cognate receptor PK2-R1. Subsequently, the activation of AstA[+] neurons then directly inhibited sweet sensation via AstA peptide and its cognate receptor AstA-R1 expressed in sweet-sensing Gr5a[+] neurons. We also showed that Neuromedin U (NMU), the mammalian homolog of fly hugin, served as an energy sensor to suppress sweet sensation. Therefore, these data identify hugin[+] neuron as a glucose-responsive central energy-sensing module that modulates sweet sensation across species.

## Introduction

Animal behavior is tightly coupled to internal metabolic states (*Ritterhoff and Tian, 2023*). Hunger profoundly alters feeding-related behaviors, enhancing sensitivity to appetitive cues and promoting food-seeking, whereas satiety actively suppresses these responses to shift behavioral priorities (*Smith et al., 2022*; *Yu et al., 2016*). Such state-dependent modulation allows animals to dynamically allocate resources to maintain energy homeostasis (*Roh and Kim, 2016*). While the neural mechanisms that drive feeding during hunger have been extensively characterized (*Tan et al., 2024*), the

central circuits that sense satiety signals—specifically circulating glucose—to directly inhibit sensory processing remain less understood.

In both mammals and insects, 'hunger signals' act as powerful accelerators for feeding. In the mammalian hypothalamus, AgRP neurons are activated by energy deficits to promote consumption (*De Solis et al., 2024*; *Sayar-Atasoy et al., 2024*), while in *Drosophila*, dopaminergic and neuropeptide F (NPF) pathways increase the gain of sweet-sensing gustatory neurons to drive sugar intake (*Chung et al., 2017*; *Shohat-Ophir et al., 2012*; *Inagaki et al., 2012*). These systems explain how animals ramp up feeding motivation when energy is low. However, to prevent overconsumption, the brain must also possess precise 'braking' mechanisms that detect elevated energy states and rapidly dampen sensory sensitivity.

Dietary sugar is a primary satiety signal (*Qi et al., 2021*), yet the pathway linking central sugar sensing to behavioral inhibition remains elusive. In *Drosophila*, high-sugar diets are known to reduce sweet sensation via gut-derived signals or through central sensors like insulin-producing cells and DH44-expressing neurons that regulate general metabolic states (*Zhao et al., 2022*; *Oh et al., 2019*; *Dus et al., 2015*). Despite these advances, a critical gap remains: identifying a direct central sensor that detects acute rises in circulating glucose and acts specifically to inhibit the peripheral sweet-sensing machinery. Hugin-expressing neurons are promising candidates for this role, as they have been implicated in satiety-related feeding regulation (*Schlegel et al., 2016*; *Schoofs et al., 2014*). Furthermore, Allatostatin A (AstA) has been proposed as a downstream signal capable of inhibiting sweet sensation (*Hergarden et al., 2012*). However, whether and how these components form a unified circuit to sense glucose and apply the 'brake' on feeding has not been established.

Here, we identify a complete neural circuitry that bridges this gap, directly linking internal glucose levels to the modulation of sweet taste sensation. We show that a specific subpopulation of hugin-expressing neurons functions as a central energy sensor, detecting elevated circulating glucose via the glucose transporter Glut1 and ATP-sensitive potassium channels. Upon activation, these neurons release hugin neuropeptide to activate downstream AstA-expressing neurons via the PK2-R1 receptor. Subsequently, AstA neurons directly inhibit sweet-sensing Gr5a$^+$ gustatory neurons through AstA-AstA-R1 signaling. This pathway functions as a glucose-responsive inhibitory module whose endogenous activity is elevated in fed flies, thereby contributing to the suppression of sweet sensitivity after feeding. Finally, we demonstrate that Neuromedin U (NMU), the mammalian homolog of hugin (*Melcher et al., 2006*), similarly acts as a central energy sensor in mice to modulate sweet-responsive circuits, revealing a conserved neural strategy for coupling metabolic state to sensory processing across species.

## Results

### hugin$^+$ neurons were a novel glucose sensor in the fly brain

Flies extend their proboscis in response to appetitive cues, a behavioral element known as the proboscis extension reflex (PER) (*Song et al., 2023*; *Marella et al., 2012*). As previously observed, hunger significantly enhances this reflex (*Marella et al., 2012*). In our study, we first replicated this phenomenon: In the state of hunger, flies exhibited elevated PER toward various concentrations of sugar compared to those under fed conditions (*Figure 1A and B*). These results confirmed the notion that sweet sensation was influenced by the internal energy state, elevated by starvation and suppressed by satiety (*Marella et al., 2012*; *Inagaki et al., 2014*). Sweet sensation in flies is mediated by sweet-sensing Gr5a$^+$ neurons located on the proboscis (*Fujii et al., 2015*). Consistent with the behavioral effect, we also confirmed that Gr5a$^+$ neurons exhibited elevated calcium responses to sugar under starvation conditions (*Figure 1—figure supplement 1A*).

Dopaminergic neurons in the subesophageal zone (SEZ) region of the fly brain acted as a hunger sensor, promoting sweet sensation via a specific dopamine receptor DopEcR expressed in Gr5a$^+$ neurons (*Marella et al., 2012*; *Inagaki et al., 2014*). However, although the gene knockout of tyrosine hydroxylase (TH) (*Budnik and White, 1987*), a key enzyme for dopamine biosynthesis, significantly suppressed sweet sensation in starved fruit flies (*Figure 1—figure supplement 1B*), these flies still showed a further reduction in sugar-induced PER under sated conditions (*Figure 1—figure supplement 1C and D*). These results suggest the existence of dopamine-independent mechanisms for sensing satiety and hunger in fruit flies that exert modulatory effects on sweet sensation.

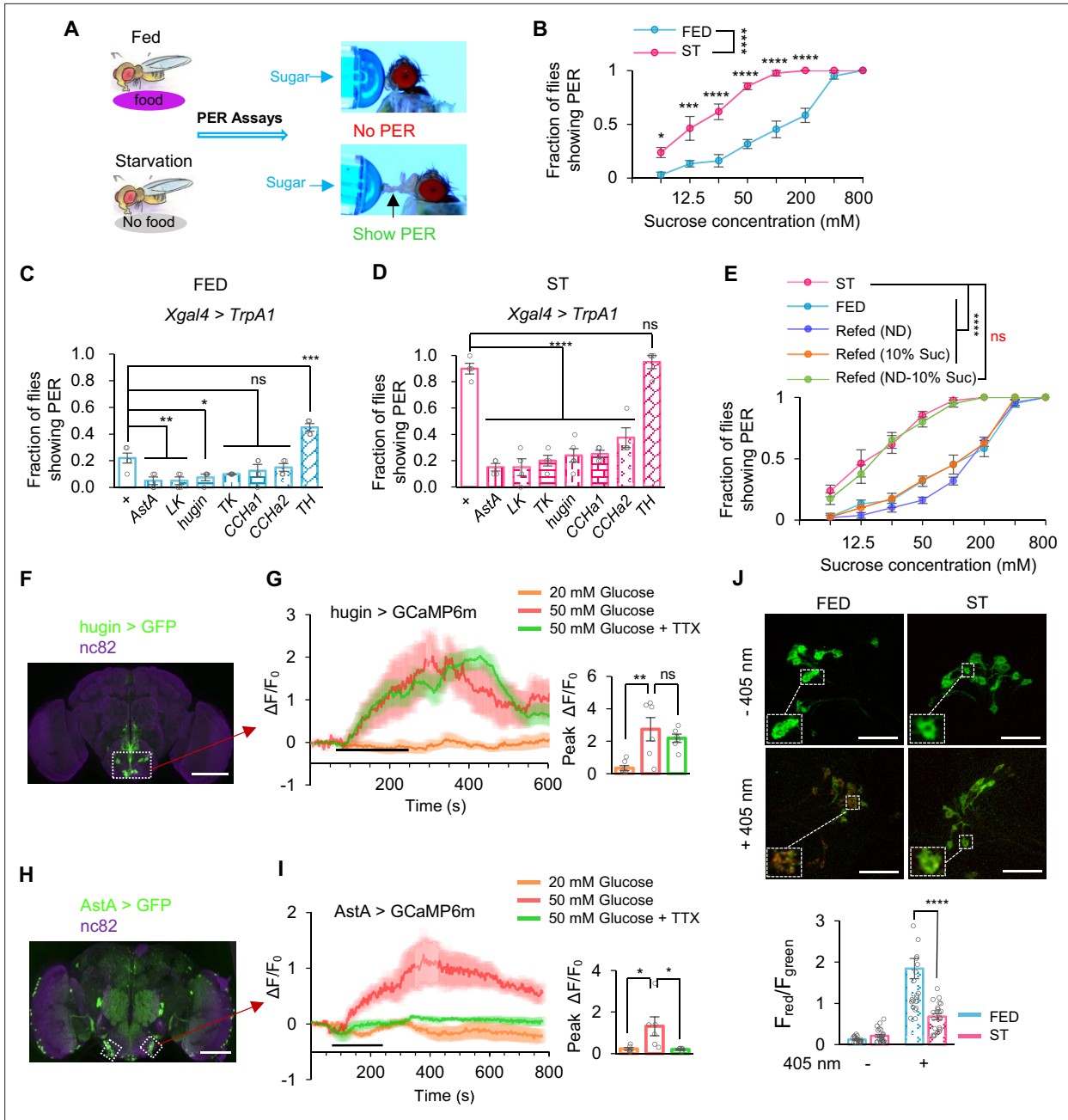

**Figure 1.** hugin⁺ neurons were a novel glucose sensor in the fly brain. (**A**) Experimental procedure schematic. Newly eclosed flies were gathered and provided with regular food for 5 days, followed by a 12 hr period of starvation. Proboscis extension reflex (PER) assays were conducted. (**B**) Fraction of flies showing PER to different concentrations of sucrose (two-way ANOVA; *, p=0.0349; ***, p=0.0002; ****, p<0.0001; n=4 groups, each of 10 flies). (**C–D**) Fraction of flies of the indicated genotypes under 30°C showing PER to 400 mM of sucrose (one-way ANOVA; *, p=0.0233; **, p=0.0054; ***, p=0.0001; ****, p<0.0001; n=4–5 groups, each of 10 flies). (**E**) Fraction of indicated flies showing PER to different concentrations of sucrose (ND represent standard fly food) (two-way ANOVA; ****, p<0.0001; n=4–5 groups, each of 10 flies). (**F**) hugin expression in the brain, illustrated by mCD8::GFP expression driven by *hugin^GAL4*. Scale bar represents 100 μm. (**G**) Representative traces and quantification of ex vivo calcium responses of hugin⁺ neurons during the perfusion of glucose with or without TTX (one-way ANOVA; **, p=0.002; n=6–7). Horizontal black bar represents the duration of the indicated glucose solution stimulation. (**H**) AstA expression in the brain, illustrated by mCD8::GFP expression driven by *AstA^GAL4*. Scale bar represents 100 μm. (**I**) Representative traces and quantification of ex vivo calcium responses of AstA⁺ neurons during the perfusion of glucose with or without TTX (one-way ANOVA; *, p=0.0187 for 50 mM glucose vs 50 mM glucose+TTX and p=0.022 for 50 mM glucose vs 20 mM glucose; n=6). (**J**) Representative images of pre-photoconversion (pre-PC) and post-photoconversion (post-PC) calcium-modulated photoactivatable ratiometric integrator (CaMPARI) signal in hugin-expressing neurons (upper). The red:green ratio represents intracellular Ca²⁺ concentrations (lower). Scale bar

*Figure 1 continued on next page*

*Figure 1 continued*

represents 100 µm (one-way ANOVA; ****, p<0.0001; n=18–23). One-way and two-way ANOVA followed by post hoc test with Bonferroni correction were used for multiple comparisons when applicable.

The online version of this article includes the following source data and figure supplement(s) for figure 1:

**Source data 1.** Source data contain numerical values and statistical results for *Figure 1*.

**Figure supplement 1.** Satiety suppressed sweet sensitivity in a dopamine-independent manner.

**Figure supplement 1—source data 1.** Source data contain numerical values and statistical results for *Figure 1—figure supplement 1*.

**Figure supplement 2.** The effect of satiety signals on taste modulation.

**Figure supplement 2—source data 1.** Source data contain numerical values and statistical results for *Figure 1—figure supplement 2*.

**Figure supplement 3.** Activation of IPCs decreased the circulating sugar levels and increased sweet sensitivity.

**Figure supplement 3—source data 1.** Source data contain numerical values and statistical results for *Figure 1—figure supplement 3*.

**Figure supplement 4.** Sugar intake suppressed sweet sensation.

**Figure supplement 4—source data 1.** Source data contain numerical values and statistical results for *Figure 1—figure supplement 4*.

**Figure supplement 5.** Starvation led to a decrease in the circulating sugar levels.

**Figure supplement 5—source data 1.** Source data contain numerical values and statistical results for *Figure 1—figure supplement 5*.

**Figure supplement 6.** The response of different neuropeptide-expressing neurons to glucose.

**Figure supplement 6—source data 1.** Source data contain numerical values and statistical results for all *Figure 1—figure supplement 6*.

**Figure supplement 7.** The calcium response of hugin neurons upon glucose stimuli.

**Figure supplement 7—source data 1.** Source data contain numerical values and statistical results for *Figure 1—figure supplement 7*.

**Figure supplement 8.** The calcium response of AstA$^+$ neurons upon glucose treatment.

**Figure supplement 8—source data 1.** Source data contain numerical values and statistical results for *Figure 1—figure supplement 8*.

In the terminating phase of feeding behavior, satiety signals play a crucial role in ceasing food consumption and ensuring an appropriate amount of food ingestion. A number of neuropeptidergic neurons were identified as putative satiety signals (*Lin et al., 2019*). Notably, some of these neurons, including those secreting neuropeptides LK, TK, hugin, AstA, CCHa1, and CCHa2, are also distributed in the SEZ region (*Deng et al., 2019*). It has been reported that the sweet-sensing gustatory neurons primarily receive regulatory inputs from neurons in the SEZ region (*Song et al., 2023*; *Marella et al., 2012*). Therefore, we hypothesized that these neuropeptidergic neurons might function as satiety signals, imposing inhibitory effects on sweet sensation after adequate food ingestion.

To test this hypothesis, we selectively expressed the temperature-sensitive ion channel TrpA1 in these neurons and activated them at 30°C, triggering the release of corresponding neuropeptides. Indeed, activation of LK-, TK-, hugin-, AstA-, CCHa1-, and CCHa2-expressing neurons robustly suppressed sweet taste sensitivity in both fed and starved flies (*Figure 1C and D*; also see *Figure 1—figure supplement 2*). Moreover, LK-, TK-, and hugin-expressing neurons exerted particularly strong inhibition under satiated conditions (*Figure 1C and D*; also see *Figure 1—figure supplement 2*). These results reveal a multilayered, state-dependent neuropeptidergic network that finely tunes gustatory processing and feeding behavior. In contrast, activating dopaminergic neurons enhanced sweet perception in sated flies as expected (*Figure 1C*). In starved flies, activating dopaminergic neurons elicited no change in PER likely due to a ceiling effect (*Figure 1D*).

In mammals and flies, circulating sugar levels are directly linked to satiety/hunger state (*Qi et al., 2021*; *Lin et al., 2019*; *Yoon and Diano, 2021*). We then asked whether changes in circulating sugar levels exerted an effect on sweet sensation. Activating IPCs in the fly brain released *Drosophila* insulin-like peptides (DILPs), the fly analog of mammalian insulin (*Brogiolo et al., 2001*), into the hemolymph and reduced circulating sugar levels (*Figure 1—figure supplement 3A*). As a result, such manipulation enhanced sweet sensitivity of fed flies (*Figure 1—figure supplement 3B and C*). Therefore, it was plausible that changes in circulating sugar might be the driver of changes in sweet sensation.

Indeed, allowing starved flies to refeed on normal fly diet or sucrose alone for 15 min both resulted in a significant decrease in sweet sensation (*Figure 1E*). Refeeding with multiple caloric sugars similarly suppressed sweet sensation (*Figure 1—figure supplement 4A*), indicating that energy intake contributes to the attenuation of sweet sensitivity. To examine whether sweet taste experience itself also plays a role, we refed starved flies with arabinose, a sweet but non-nutritive sugar. Arabinose

feeding likewise led to a reduction in sweet sensitivity (*Figure 1—figure supplement 4B*), suggesting that sweet taste experience can contribute to the suppression of sweet perception, potentially through sensory adaptation. Consistent with this interpretation, responses to other sweet compounds, including trehalose and fructose, were also reduced following sucrose refeeding (*Figure 1—figure supplement 4C and D*). Conversely, when starved flies were supplied with fly diet deprived of sucrose (but with normal levels of polysaccharides, proteins, and lipids), there was no immediate change in sweet sensation after refeeding (*Figure 1E*). These results demonstrate that sugar intake inhibits sweet sensation, probably via increasing circulating sugar levels.

Collectively, we thus hypothesized that the above neuropeptidergic neurons might directly sense an increase of circulating sugar levels and suppress sweet sensation in response. If so, these neurons should be activated by glucose, the main species of circulating sugars associated with feeding behavior in flies (*Ugrankar et al., 2018*), at the concentrations resembling sated state (>50 mM in fed flies *Figure 1—figure supplement 5*). We thus conducted calcium imaging experiments in fly brains in an ex vivo preparation (*Qi et al., 2021*). Among all the six groups of neuropeptidergic neurons, we examined in the SEZ, only hugin[+] neurons and AstA[+] neurons could be activated by 50 mM glucose (*Figure 1F–I*, also see *Figure 1—figure supplement 6*). Remarkably, each of these populations subdivided into anatomically distinct SEZ subclusters (*Figure 1—figure supplement 7A*; *Figure 1—figure supplement 8A*), yet glucose-evoked $Ca^{2+}$ responses occurred exclusively within a single subcluster (*Figure 1H and I*; *Figure 1—figure supplement 7B*; *Figure 1—figure supplement 8B*). This selective activation highlighted profound functional heterogeneity in nutrient sensing among these neuron groups. When direct synaptic transmission was blocked with TTX, hugin[+] neurons, but not AstA[+] neurons, still exhibited calcium responses toward 50 mM of glucose (*Figure 1G*, green), suggesting that hugin[+] neurons are a direct glucose sensor. Furthermore, the level of circulating glucose dropped to ~20 mM in starved flies (*Figure 1—figure supplement 5*). We found that 20 mM of glucose did not activate hugin[+] and AstA[+] neurons (*Figure 1G and I*, orange), suggesting that these neurons can only be activated under sated conditions.

We also assessed the activity of hugin[+] neurons in vivo using calcium-modulated photoactivatable ratiometric integrator (CaMPARI), a method employing a photoconvertible fluorescent protein that allows the imaging of integrated calcium activity (*Fosque et al., 2015*), under both sated and starved conditions. In the CaMPARI assay, when neurons are activated, leading to an increase in free calcium ions, exposure to 405 nm light induces a permanent green-to-red photoconversion (PC) (*Fosque et al., 2015*). We found an increased ratio of red/green fluorescence in hugin[+] neurons in the sated state compared to that upon starvation (*Figure 1J*). These findings strongly imply that hugin[+] neurons function as a glucose-sensitive energy sensor whose activity tracks internal metabolic state.

## Glucose activated hugin[+] neurons via an ATP-sensitive potassium channel

We then investigated the cellular mechanism through which glucose could directly activate hugin[+] neurons. In both mammals and flies, extracellular glucose is usually transported into the cytosol to activate glucose-sensitive neurons (*Oh et al., 2019*; *Dus et al., 2015*; *Parton et al., 2007*). When fly brains were pre-treated with phlorizin, a blocker of glucose transport (*Oh et al., 2019*; *Dus et al., 2015*), hugin[+] neurons showed diminished calcium responses to glucose, suggesting activation of hugin[+] neurons by glucose requires intracellular glucose (*Figure 2A*).

The transport of glucose mediated by glucose transporter 1 (Glut1) can activate certain neurons in the fruit fly brain (*Oh et al., 2019*; *Dus et al., 2015*). By specifically knocking down Glut1 expression in hugin[+] neurons, we observed a significant reduction in calcium responses to glucose (*Figure 2B*). Consistently, these flies showed a slight yet significant increase in their PER responses to glucose (*Figure 2C*). Again, these results further confirm that glucose needs to be transported into hugin[+] neurons to modulate neuronal activity and behavioral output.

In pancreatic β-cells and certain fly neurons (*Oh et al., 2019*; *Parton et al., 2007*; *Wang et al., 2022*), intracellular glucose modulates cell excitability primarily through its effect on intracellular energy metabolism and ATP-sensitive potassium channels ($K_{ATP}$). We pre-treated fly brains with a $K_{ATP}$ inhibitor (glibenclamide) and observed a significant calcium response in hugin[+] neurons (*Figure 2D*). Hexokinase, a key enzyme in glucose metabolism that generates ATP, also plays a crucial role in this process. When hexokinase activity was restricted by alloxan, glucose-induced calcium response in

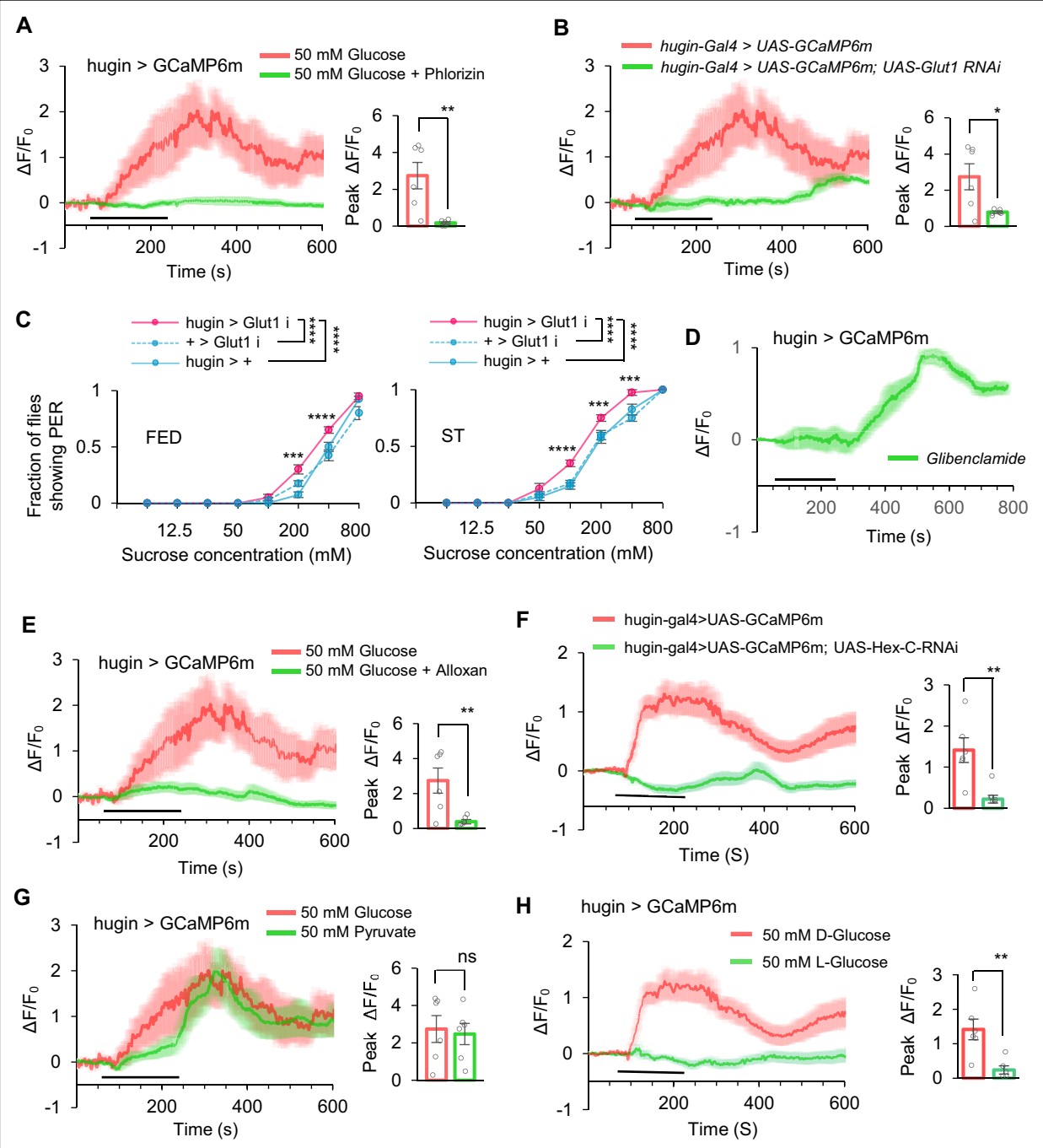

**Figure 2.** Glucose activated hugin+ neurons via Gltu1 and K_ATP channel. (**A**) Representative traces and quantification of ex vivo calcium responses of hugin+ neurons during the perfusion of glucose with or without phlorizin (1 mM), (*t*-test; **, p=0.0052; n=6–7). Horizontal black bar represents the duration of indicated glucose solution stimulation. (**B**) Representative traces and quantification of ex vivo calcium responses of hugin+ neurons in indicated flies during the perfusion of glucose (*t*-test; *, p=0.022; n=6). (**C**) Fraction of flies of the indicated genotypes showing proboscis extension reflex (PER) to different concentrations of sucrose (two-way ANOVA; ***, p=0.0009 for FED 200 mM and 0.0001 for ST 200 mM and 400 mM; ****, p<0.0001; n=4 groups, each of 10 flies). (**D–E**) Representative traces and quantification of ex vivo calcium responses of hugin+ neurons during the perfusion of glibenclamide (100 μM, **D**), glucose with alloxan (10 μM, **E**) (*t*-test; **, p=0.0091; n=6). (**F**) Representative traces and quantification of ex vivo calcium responses of hugin+ neurons in indicated flies during the perfusion of glucose (*t*-test; **, p=0.0011; n=6). (**G**) Representative traces and quantification of ex vivo calcium responses of hugin+ neurons during the perfusion of pyruvate (50 mM), (*t*-test; ns, p>0.05; n=6). (**H**) Representative traces and quantification of ex vivo calcium responses of hugin+ neurons during the perfusion of D-glucose and L-glucose (50 mM), (*t*-test; **, p=0.0045; n=6). Student's *t*-test and two-way ANOVA followed by post hoc test with Bonferroni correction were used for multiple comparisons when applicable.

The online version of this article includes the following source data for figure 2:

**Source data 1.** Source data contain numerical values and statistical results for *Figure 2*.

hugin$^+$ neurons was significantly suppressed (*Figure 2E*). Similarly, when hexokinase was specifically knocked down in hugin$^+$ neurons, a comparable reduction in glucose-induced calcium responses was observed (*Figure 2F*). Pyruvate, the end product from glycolysis, can lead to ATP generation via the tricarboxylic acid cycle. We also found that pyruvate stimulation led to a significant calcium response in hugin$^+$ neurons (*Figure 2G*). As a specificity control, nonmetabolizable L-glucose failed to activate hugin$^+$ neurons (*Figure 2H*).

Collectively, these findings suggest that extracellular glucose directly enters hugin$^+$ neurons through specific glucose transporter and subsequently modulates the activity of $K_{ATP}$ through ATP production, ultimately leading to neuronal depolarization. We note that blocking glucose transport or metabolism could, in principle, nonspecifically affect neuronal function, but the rapid and selective nature of the observed responses argues against a generalized loss of excitability.

## AstA$^+$ neurons were direct downstream targets of hugin$^+$ neurons

Since hugin$^+$ neurons were directly involved in glucose sensing and regulated sweet sensation (*Figure 1G* and *Figure 3A and B*), we sought to identify their downstream target. We then asked whether the hugin peptide had a direct effect on sweet sensation. Upon microinjection of synthetic hugin peptide, we observed a notable reduction in sweet sensation under both sated and starved states (*Figure 3C*). In line with this observation, following the injection of hugin peptide, we observed a significant reduction of sugar-induced calcium responses in sweet-sensing Gr5a$^+$ neurons (*Figure 3D*). Thus, the inhibitory function of hugin$^+$ neurons on sweet sensation was likely mediated by the secretion of hugin peptide.

There were two hugin receptors, PK2-R1 and PK2-R2, in fruit flies (*Park et al., 2002*). We knocked down their expression by RNAi in the nervous system and then tested flies' PER to sugar. Knocking down PK2-R1 but not PK2-R2 significantly enhanced sweet perception in both sated and starved states (*Figure 3E and F*), suggesting a crucial role of PK2-R1 in receiving the input of hugin$^+$ neurons in satiety sensing. To validate this finding, we further tested PK2-R1 null mutants and observed a similar enhancement in sweet sensitivity (*Figure 3—figure supplement 1*). Although PK2-R2 knockdown did not enhance PER, we observed a modest but reproducible reduction in PER in starved flies (*Figure 3F*). This suggests potential functional heterogeneity within hugin signaling pathways. Given the anatomical diversity of hugin neurons, distinct subpopulations may engage different receptor-dependent mechanisms that exert complementary or even opposing effects on feeding behavior. Further dissection of PK2-R2-dependent signaling will be required to clarify this complexity.

Interestingly, we did not observe any expression of PK2-R1 in the proboscis where gustatory sensory neurons were distributed, indicating that sweet-sensing Gr5a$^+$ neurons could not directly receive input from hugin$^+$ neurons via PK2-R1 (*Figure 3—figure supplement 2*). AstA$^+$ neurons served as a plausible relay between hugin$^+$ neurons and sweet-sensing Gr5a$^+$ neurons (*Figure 4A and B*), especially based on a clear co-expression of hugin receptor PK2-R1 and AstA in the SEZ region of the fly brain (*Figure 4C*). Notably, these double-labeled neurons corresponded precisely to the AstA subcluster that exhibited glucose-specific Ca$^{2+}$ responses (*Figure 1H*). We also used optogenetic tools to further confirm that hugin$^+$ neurons acted upstream of AstA$^+$ neurons. We expressed the light-sensitive neuronal activator CsChrimson in hugin$^+$ neurons and calcium indicator GCaMP6m in AstA$^+$ neurons, respectively (*Klapoetke et al., 2014*). As shown in *Figure 4D*, opto-activation of hugin$^+$ neurons led to a robust calcium transient in AstA$^+$ neurons.

We then investigated whether the hugin-AstA circuitry affected sweet sensation. As expected, knocking down PK2-R1 (but not PK2-R2) in AstA$^+$ neurons led to a significant increase in sugar-induced PER (*Figure 4E and F*). Therefore, AstA$^+$ neurons are the direct downstream target of hugin$^+$ neurons in satiety sensing. hugin$^+$ neurons can directly sense elevated circulating glucose levels, promote the release of neuropeptide hugin, and then activate AstA$^+$ neurons via its cognate receptor PK2-R1.

hugin$^+$ neurons are distributed in both the brain and the ventral nerve cord (VNC). The hugin$^+$ neurons in the VNC sense sugar and inhibit feeding behavior by suppressing the activity of DH44 neurons (*Oh et al., 2021*). To determine which hugin$^+$ neurons regulate the activity of AstA neurons and sweet sensation, we performed the following experiments: First, we physically severed the neural connection between the brain and the VNC. Under this condition, activation of hugin$^+$ neurons still significantly reduced PER responses (*Figure 4—figure supplement 1A and B*), indicating that hugin$^+$ neurons within the brain are sufficient to modulate sweet perception. Second, in optogenetic

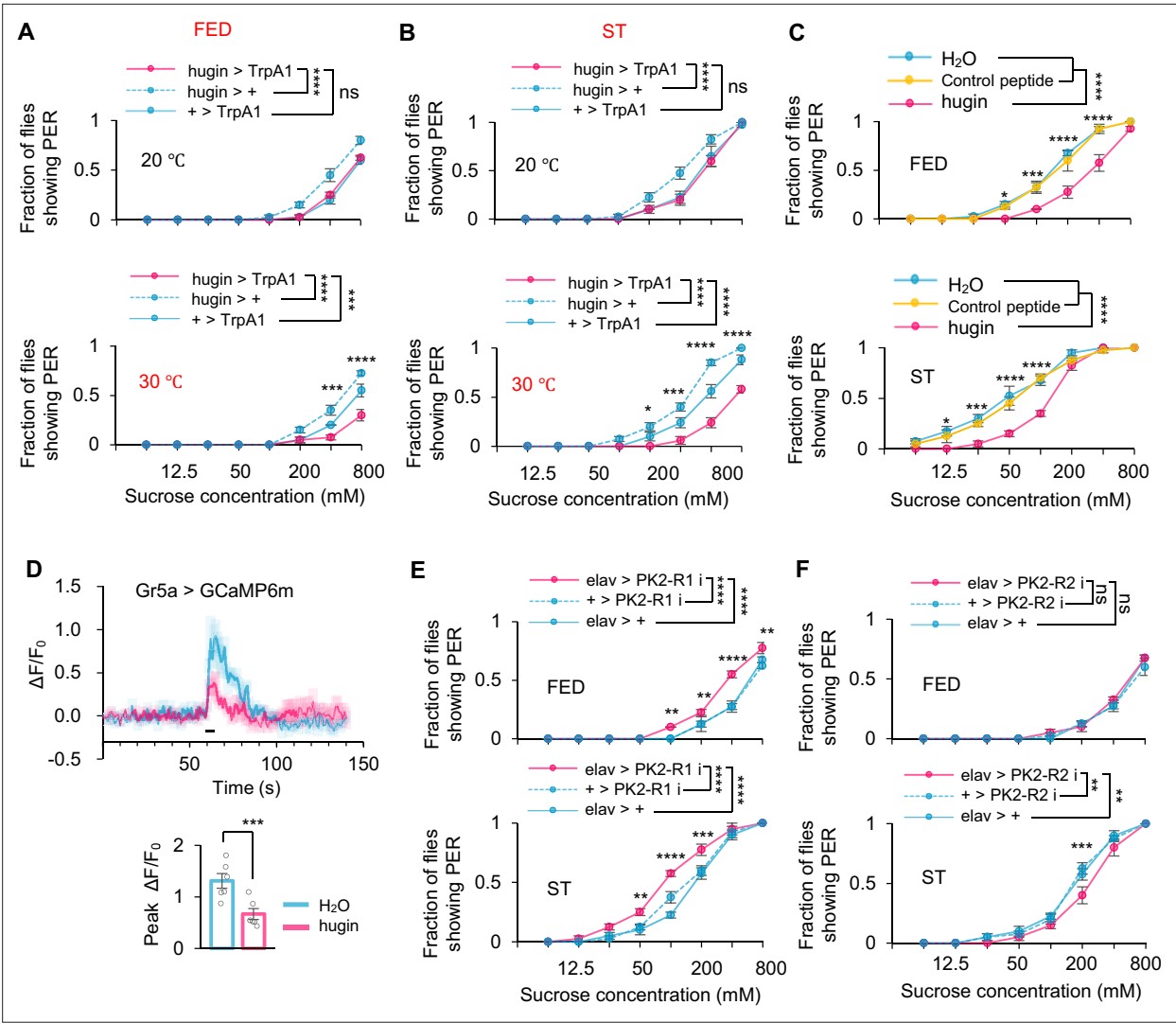

**Figure 3.** hugin+ neurons suppressed sweet taste through PK2-R1. (**A–B**) Fraction of flies of the indicated genotypes and environmental temperatures showing proboscis extension reflex (PER) to different concentrations of sucrose (two-way ANOVA; *, p=0.043; ***, p=0.0004 for curve FED hugin>TrpA1 vs +>TrpA1 at 30°C, p=0.0009 for FED 400 mM at 30°C and p=0.0001 for ST 200 mM at 30°C; ****, p<0.0001; n=4–5 groups, each of 10 flies). (**C**) Fraction of flies showing PER to different concentrations of sucrose (two-way ANOVA; *, p=0.0486 for FED 50 mM and p=0.0438 for ST 12.5 mM; ***, p=0.0002 for FED 100 mM and p=0.0007 for ST 25 mM; ****, p<0.0001; n=4 groups, each of 10 flies). Flies were injected with saline or synthetic hugin for 30 min before the assay. (**D**) Representative traces (upper) and quantification (lower) of peak calcium transients of Gr5a+ neurons in indicated flies upon 5% sucrose after injection of synthetic hugin (t-test; ***, p=0.0047; n=6). Horizontal black bar represents the duration of sucrose stimulation. (**E–F**) Fraction of flies of the indicated genotypes showing PER to different concentrations of sucrose (two-way ANOVA; **, p=0.0055 for figure E and p=0.003 and 0.0016 for curve elav>PK2R2i vs +>PK2R2i and elav >PK2R2i v elav > +; ***,p=0.0002 for figure E ST 200 mM and p=0.0005 for figure F ST 200 mM; ****, p<0.0001; n=4 groups, each of 10 flies). Two-way ANOVA followed by post hoc test with Bonferroni correction was used for multiple comparisons when applicable.

The online version of this article includes the following source data and figure supplement(s) for figure 3:

**Source data 1.** Source data contain numerical values and statistical results for *Figure 3*.

**Figure supplement 1.** Knockout of *PK2-R1* enhanced sweet sensation.

**Figure supplement 1—source data 1.** Source data contain numerical values and statistical results for *Figure 3—figure supplement 1*.

**Figure supplement 2.** The expressing pattern of PK2-R1 in the proboscis.

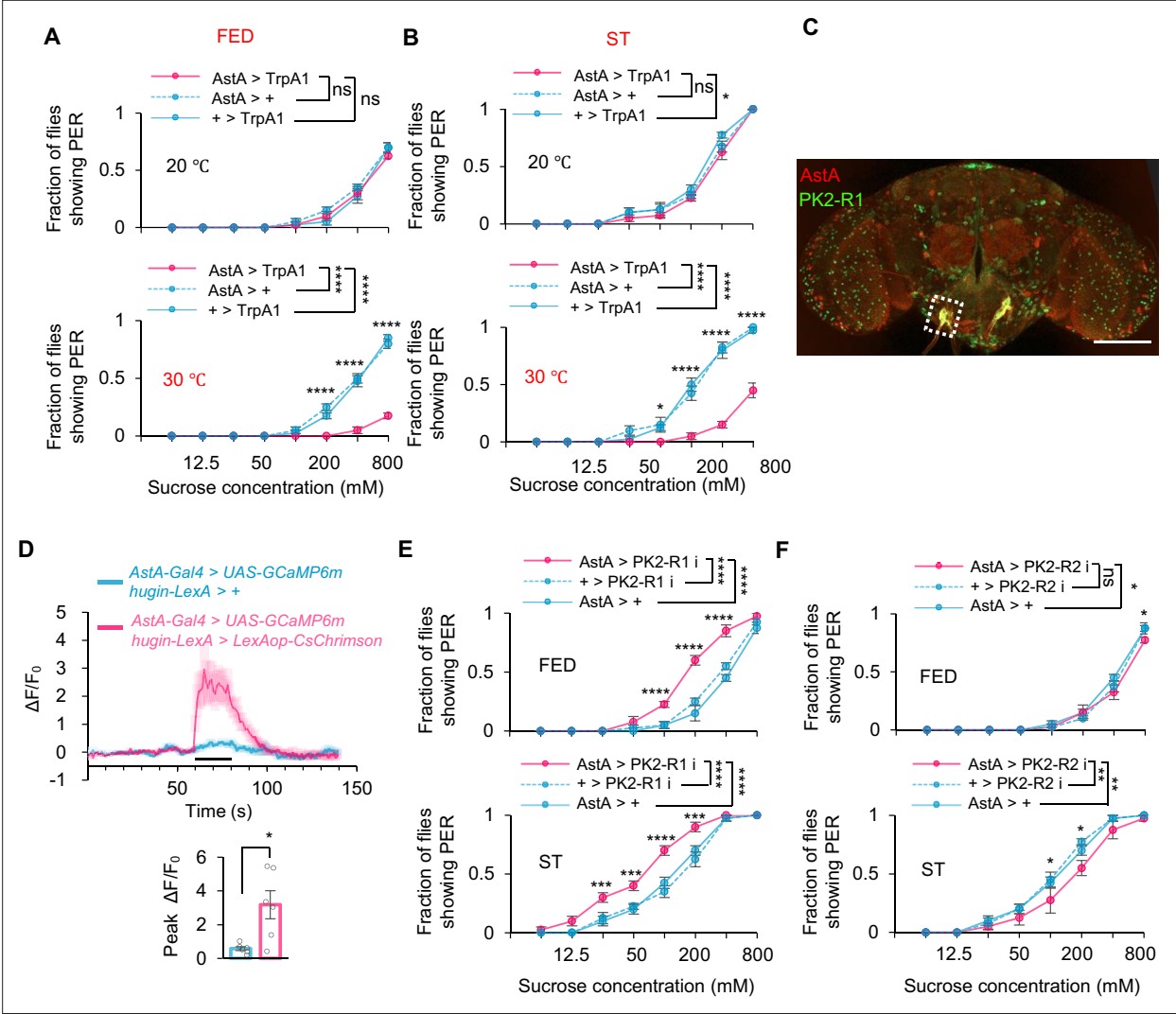

**Figure 4.** hugin+ neurons activated AstA+ neurons through PK2-R1. (**A–B**) Fraction of flies of the indicated genotypes and environmental temperatures showing proboscis extension reflex (PER) to different concentrations of sucrose (two-way ANOVA; *, p=0.0153 for curve ST AstA >TrpA1 vs +>TrpA1 at 20°C and p=0.0237 for (**B**) 50 mM at 30°C; ****, p<0.0001; n=4–5 groups, each of 10 flies). (**C**) Co-localization (dashed box) of PK2-R1+ neurons (green) and AstA+ neurons (red) in the subesophageal zone (SEZ) region. Scale bar represents 100 μm. (**D**) Representative traces (upper) and quantification (lower) of peak calcium transients of AstA+ neurons after the photoactivation of hugin+ neurons (t-test; *, p=0.0114; n=6) from in vivo calcium imaging. Horizontal black bar represents the duration of red-light stimulation. (**E–F**) Fraction of flies of the indicated genotypes showing PER to different concentrations of sucrose (two-way ANOVA; *, p=0.0486 for curve FED AstA>PK2-R2i vs AstA >+ and p=0.0024 for FED 800 mM and p=0.033 for ST 100 mM and 200 mM; **, p=0.0043 and 0.0011 for AstA>PK2 R2 vs AstA >+ and +>PK2-R2i; ***, p=0.0008 for 25 mM and 50 mM and p=0.0001 for 200 mM; ****, p<0.0001; n=4 groups, each of 10 flies). Student's t-test and two-way ANOVA followed by post hoc test with Bonferroni correction were used for multiple comparisons when applicable.

The online version of this article includes the following source data and figure supplement(s) for figure 4:

**Source data 1.** Source data contain numerical values and statistical results for *Figure 4*.

**Figure supplement 1.** hugin+ neurons in the brain activated AstA+ neurons.

**Figure supplement 1—source data 1.** Source data contain numerical values and statistical result *Figure 4—figure supplement 1*.

experiments using isolated brain preparations, light stimulation of hugin+ neurons elicited robust calcium responses in AstA+ neurons (*Figure 4—figure supplement 1C*), demonstrating a direct functional coupling between brain hugin+ neurons and AstA+ neurons.

Notably, comparison of intact and severed preparations revealed that the magnitude of AstA+ neuronal activation was greater when VNC neurons were preserved, suggesting that VNC hugin+ neurons may provide additional modulatory input that enhances AstA activation. Together, these

results indicate that hugin$^+$ neurons in the SEZ constitute a primary pathway linking sugar sensing to AstA activation and PER regulation, while VNC hugin$^+$ neurons contribute to the overall strength of hugin signaling.

## AstA$^+$ neurons directly inhibited Gr5a$^+$ neurons

Activation of AstA$^+$ neurons suppressed sweet sensation, resembling the effect of activating hugin$^+$ neurons (*Hergarden et al., 2012*; *Figure 4A and B*). We also evaluated the behavioral effect of AstA peptide. Similar to hugin, the microinjection of AstA resulted in a reduction in sweet sensitivity in fruit flies (*Figure 5A*) and the activity of Gr5a$^+$ neurons (*Figure 5B*). We then speculated whether AstA$^+$ neurons and AstA peptide might function directly on Gr5a$^+$ neurons. The cell bodies of sweet-sensing Gr5a$^+$ neurons are located in the proboscis, whereas their axons extend to the SEZ region of the fly brain (*Dahanukar et al., 2007*). We observed distributions of the projections of AstA$^+$ neurons and Gr5a$^+$ neurons in the SEZ region (*Figure 5C*), suggesting that Gr5a$^+$ neurons might directly receive AstA signals in this region.

AstA has two cognate receptors in fruit flies, AstA-R1 and AstA-R2 (*Larsen et al., 2001*). Co-labeling experiments in the proboscis confirmed the expression of AstA-R1 in Gr5a$^+$ neurons (*Figure 5D*, arrows), while AstA-R2 was not expressed in the proboscis (*Figure 5—figure supplement 1*). Functionally, knockdown of AstA-R1 specifically in Gr5a$^+$ neurons, but not that of AstA-R2, significantly enhanced sweet sensation of flies (*Figure 5E and F*). Similar results were observed in AstA-R1 knockout flies (*Figure 5—figure supplement 2*).

Using optogenetic tools, we also confirmed the function of hugin$^+$ neurons and AstA$^+$ neurons on Gr5a$^+$ neurons. Activating hugin$^+$ neurons and AstA$^+$ neurons both significantly reduced sugar-induced responses of Gr5a$^+$ neurons (*Figure 5G and H*). Collectively, these results suggest that satiety-sensing hugin-AstA circuitry can directly modulate sweet sensation of flies via suppressing the activity of Gr5a$^+$ gustatory neurons.

## hugin-AstA circuitry suppressed food consumption

Collectively, the hugin-AstA circuitry senses satiety signals and inhibits sweet sensation. Considering the crucial role of sweet sensation in regulating feeding behavior, we hypothesized that hugin-AstA circuitry was also involved in the regulation of food consumption.

We expressed a temperature-sensitive blocker of synaptic transmission (Shibire$^{ts1}$) in hugin$^+$ neurons or AstA$^+$ neurons (*Kitamoto, 2001*) and assayed the feeding behavior of flies under both permissive temperature (*Figure 6A*, blue) and non-permissive temperature (*Figure 6A*, red). In contrast to the suppression of sweet sensitivity observed upon thermogenetic activation of these neurons, silencing hugin$^+$ neurons led to enhanced sweet-driven feeding behavior (*Figure 6B*). Consistently, inhibiting synaptic transmission in either hugin$^+$ or AstA$^+$ neurons resulted in a significant increase in food consumption at the non-permissive temperature compared to controls (*Figure 6C and D*).

We also examined the role of hugin and AstA receptors in feeding behavior by testing *PK2-R1* and *AstA-R1* gene knockout flies. The results showed that knockout of both receptors enhanced feeding behavior (*Figure 6—figure supplement 1A*), consistent with increased sweet perception upon genetic manipulation of these two receptors (*Figures 4 and 5*). Moreover, these receptor knockout flies exhibited increased lipid storage, which was likely a consequence of their enhanced feeding behavior (*Figure 6—figure supplement 1B*). Additionally, we knocked down the expression of AstA-R1 in sweet-sensing Gr5a$^+$ neurons and found that these flies exhibited a significant increase in food consumption (*Figure 6—figure supplement 1C*), also likely due to the enhanced sweet sensation (*Figure 5E*).

## Mammalian homolog of hugin was also a putative satiety sensor

Our present work identified hugin-AstA circuitry as a novel satiety sensor and feeding suppressor in fruit flies. Fruit flies and mammals share highly conserved regulatory elements on feeding behavior and energy homeostasis (*Song et al., 2023*; *Stocker, 2004*; *Yarmolinsky et al., 2009*). We thus asked whether this novel satiety-sensing mechanism was also conserved in the mammalian system.

In rodent models, NMU, an analog of hugin, was also found to be an appetite suppressor (*Schlegel et al., 2016*; *Melcher et al., 2006*). We found that synthetic NMU could suppress sweet perception when injected into flies' thorax (*Figure 7—figure supplement 1*), highlighting the analogy of fly

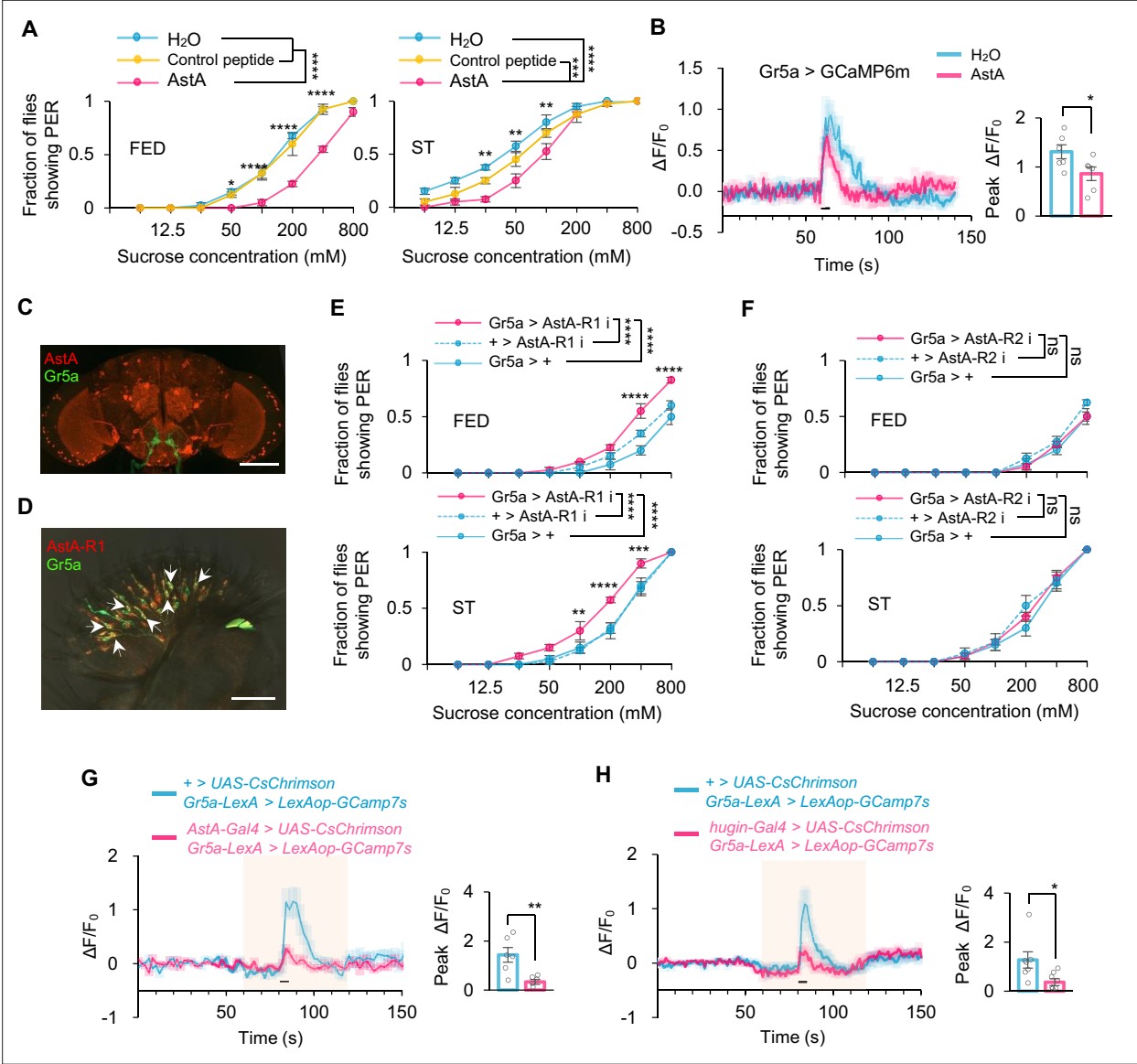

**Figure 5.** AstA[+] neurons inhibited Gr5a[+] neurons through AstA-R1. (**A**) Fraction of flies showing proboscis extension reflex (PER) to different concentrations of sucrose (two-way ANOVA; *, p=0.0219 for 50 mM; **, p=0.0072 for 25 mM and 100 mM and p=0.002 for 50 mM; ***, p=0.0002 for curve ST control peptide for AstA; ****, p<0.0001; n=4 groups, each of 10 flies). Flies were injected with saline or synthetic AstA for 30 min before the assay. (**B**) Representative traces (left) and quantification (right) of peak calcium transients of Gr5a[+] neurons in indicated flies upon 5% sucrose after injection of synthetic AstA (t-test; *, p=0.0476; n=6). Horizontal black bar represents the duration of sucrose stimulation. (**C**) Localization of Gr5a[+] neurons (green) and AstA[+] neurons (red) in the brain. Scale bar represents 100 μm. (**D**) Colocalization of Gr5a[+] neurons (green) and AstA-R1[+] neurons (red). Arrows point to the neurons co-expressing the two receptors. Scale bar represents 50 μm. (**E–F**) Fraction of flies of the indicated genotypes showing PER to different concentrations of sucrose (two-way ANOVA; **, p=0.0088; ***, p=0.0004; ****, p<0.0001; n=4 groups, each of 10 flies). (**G–H**) Representative traces (left) and quantification (right) of peak calcium transients of Gr5a[+] neurons in indicated flies upon 5% sucrose during the process of photoactivation of AstA[+] neurons (G, n=6; t-test; **, p=0.0058) and hugin[+] neurons (H, n=7–8; t-test; *, p=0.0351). Shadows represent the duration of red-light stimulation. Horizontal black bar represents the duration of sucrose stimulation. Student's t-test and two-way ANOVA followed by post hoc test with Bonferroni correction were used for multiple comparisons when applicable.

The online version of this article includes the following source data and figure supplement(s) for figure 5:

**Source data 1.** Source data contain numerical values and statistical results for *Figure 5*.

**Figure supplement 1.** The expressing pattern of AstA-R2 in the proboscis.

**Figure supplement 2.** Manipulating gene *AstA-R1* enhances proboscis extension reflex (PER).

**Figure supplement 2—source data 1.** Source data contain numerical values and statistical results for *Figure 5—figure supplement 2*.

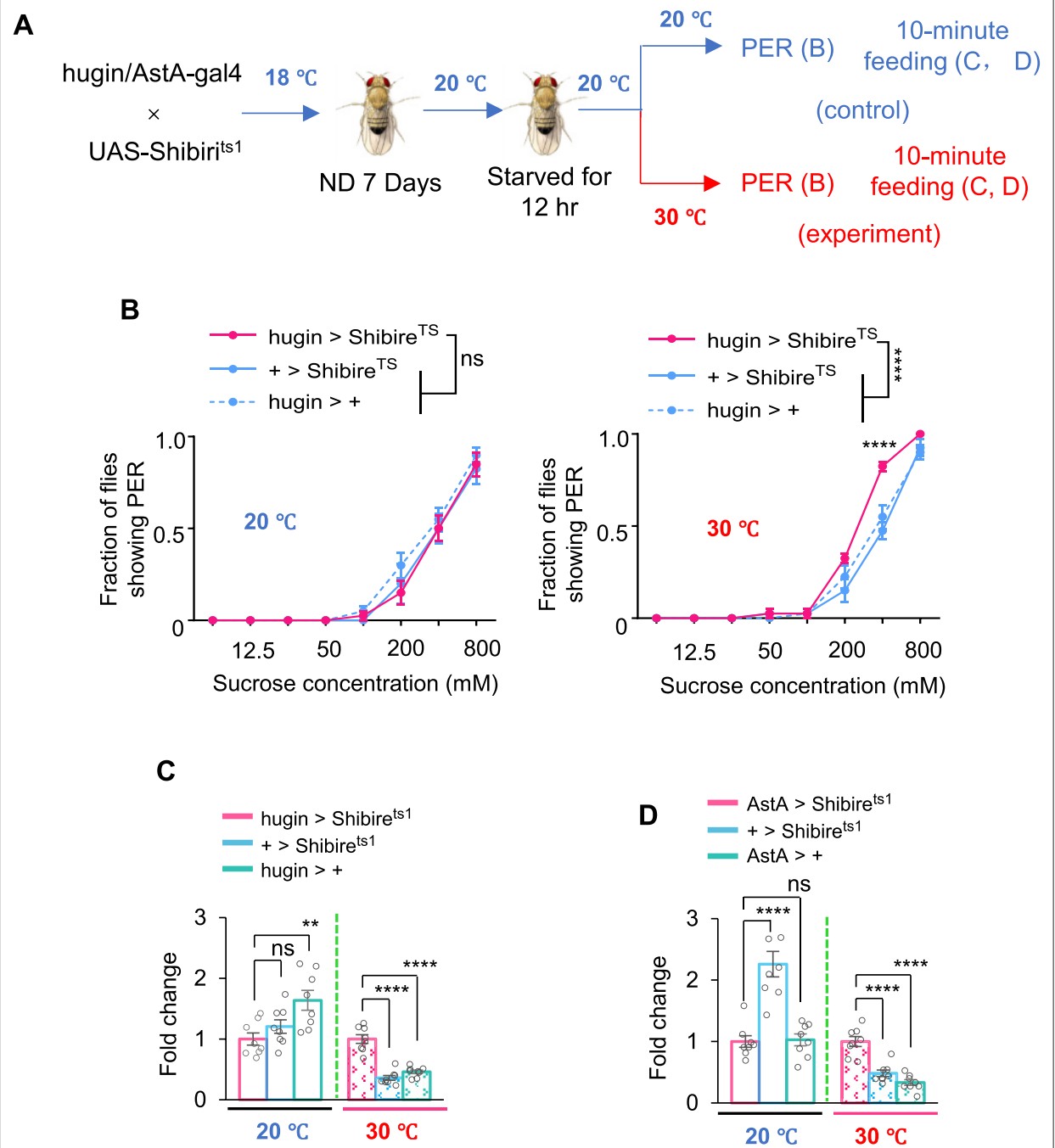

**Figure 6.** hugin-AstA circuitry inhibited feeding behavior. (**A**) The illustration of experimental design in this figure. Note that all flies were maintained at 20°C before the feeding assay to prevent neuronal silencing by Shibire[ts1] during these procedures. (**B**) Fraction of flies of the indicated genotypes and environmental temperatures showing proboscis extension reflex (PER) to different concentrations of sucrose (two-way ANOVA; ***, p=0.0003; ****, p<0.0001; n=4 groups, each of 10 flies). (**C–D**) Food consumption of flies of the indicated genotypes and environmental temperatures (one-way ANOVA; **, p=0.004; ****, p<0.0001; n=8 groups, each of 4 flies). Food consumption is presented as fold change relative to the corresponding genetic control.

The online version of this article includes the following source data and figure supplement(s) for figure 6:

**Source data 1.** Source data contain numerical values and statistical results for all *Figure 6*.

**Figure supplement 1.** hugin-AstA-Gr5a circuitry inhibited feeding behavior.

**Figure supplement 1—source data 1.** Source data contain numerical values and statistical results for *Figure 6—figure supplement 1*.

hugin and mammalian NMU. In mouse models, we observed a significant increase in NMU levels in the blood following glucose feeding (*Figure 7A*). Using the sucrose preference test to measure sweet preference, we noted that NMU injection reduced sugar preference in mice, phenocopying the effect of glucose feeding (*Figure 7B and C*). Conversely, NMU knockout enhanced sweet sensation in mice (*Figure 7D*). These data demonstrated that both *Nmu* gene and protein play a crucial role in the regulation of sweet sensation.

Neuropeptide NMU is secreted by a group of neurons located in the ventromedial hypothalamus (VMH) region of the hypothalamus (*Melcher et al., 2006*). To investigate whether NMU+ neurons functioned as an energy sensor in the mouse brain, we injected AAV-EF1α-DIO-GCaMP6m-WPRE into the VMH of NMU-Cre mice, specifically expressing GCaMP6m in NMU+ neurons. We then monitored the dynamic activity of VMH NMU neurons during glucose stimulation by using fiber photometry (*Figure 7E*). As expected, these neurons showed significant calcium responses following glucose administration in mice (*Figure 7F and G*). To confirm the specificity of GCaMP6m expression in NMU+ neurons, we co-expressed a Cre-dependent mCherry reporter in the VMH of NMU-Cre mice and performed fluorescence in situ hybridization for *Nmu* mRNA. Colocalization analysis demonstrated that GCaMP6m fluorescence was restricted exclusively to NMU+ cells (*Figure 7—figure supplement 2A*).

Next, we conducted calcium imaging of brain slices in vitro to further assess these neurons' ability to sense glucose (*Figure 7H*). Similar to hugin neurons in the fly brain, VMH$^{NMU}$ neurons in the mouse brain exhibited strong calcium responses to glucose stimulation, which could not be inhibited by TTX (*Figure 7I*), indicating their direct glucose-sensing capability. Also similar to flies, glucose activated these mouse neurons via entering the cells and modulating K$_{ATP}$ channels, as evidenced by the inhibition of calcium responses by alloxan and phlorizin (*Figure 7J*). While supraphysiological glucose concentrations were used in slice recordings to ensure reliable neuronal activation under ex vivo conditions, complementary in vivo recordings demonstrate that VMH NMU+ neurons are engaged by physiological sugar ingestion.

Calbindin 2-positive neurons (Calb2) in the rostral nucleus of the solitary tract (rNST) selectively respond to sweet tastants, facilitating sweet signal from the periphery into the brain (*Jin et al., 2021*). Because attenuated sweet sensation should correspondingly diminish central sweet-signal transmission, we asked whether NMU modulates this relay. To assess potential anatomical connectivity between VMH NMU+ neurons and rNST Calb2+ neurons, we employed a dual-virus tracing strategy. In NMU-Cre mice, we injected AAV9-EF1α-DIO-EGFP into the VMH, which revealed dense GFP+ projections in close apposition to Calb2+ neurons in the rNST (*Figure 7—figure supplement 2B*). To further confirm this projection is monosynaptic, we combined anterograde tracing by co-injecting AAV2/1-EF1α-DIO-EGFP into the VMH and AAV2/1-hsyn-SV40-NLS-Cre into the rNST. This approach yielded GFP expression in rNST Calb2+ neurons, supporting a direct VMH→rNST (NMU+→Calb2+) synaptic link (*Figure 7K*).

To evaluate the functional consequence of NMU on Calb2+ neuronal activity during sweet stimulation, we delivered AAVs encoding Cre-dependent GCaMP6m into the rNST of Calb2-Cre mice, thereby enabling real-time calcium imaging in Calb2+ neurons (*Figure 7—figure supplement 2C*). Using fiber photometry, we recorded calcium responses to sugar stimuli, both with and without NMU injection (*Figure 7L*). We found that NMU injection elicited a significant, albeit partial, suppression of calcium responses to sugar (*Figure 7M*). Although the magnitude of this reduction was modest compared to the complete silencing of activity, the effect was statistically robust and aligned with the behavioral phenotypes, indicating that NMU acts to dampen sweet taste perception during feeding.

These findings indicate that NMU+ neurons can indeed be a putative satiety sensor in the mouse brain that suppresses sweet perception, shedding light on understanding the conserved neural mechanism of feeding regulation in mammals, including humans.

## Discussion

In this study, we identify a neural pathway that directly couples internal glucose levels to the regulation of sweet taste perception and feeding behavior. In *Drosophila*, circulating glucose activates a specific subset of hugin neurons, which signal to AstA neurons to suppress the activity of peripheral sweet-sensing Gr5a neurons, thereby reducing sweet sensitivity and terminating feeding. This pathway establishes a direct neural link between internal energy state and early sensory processing (*Figure 7—figure supplement 3*).

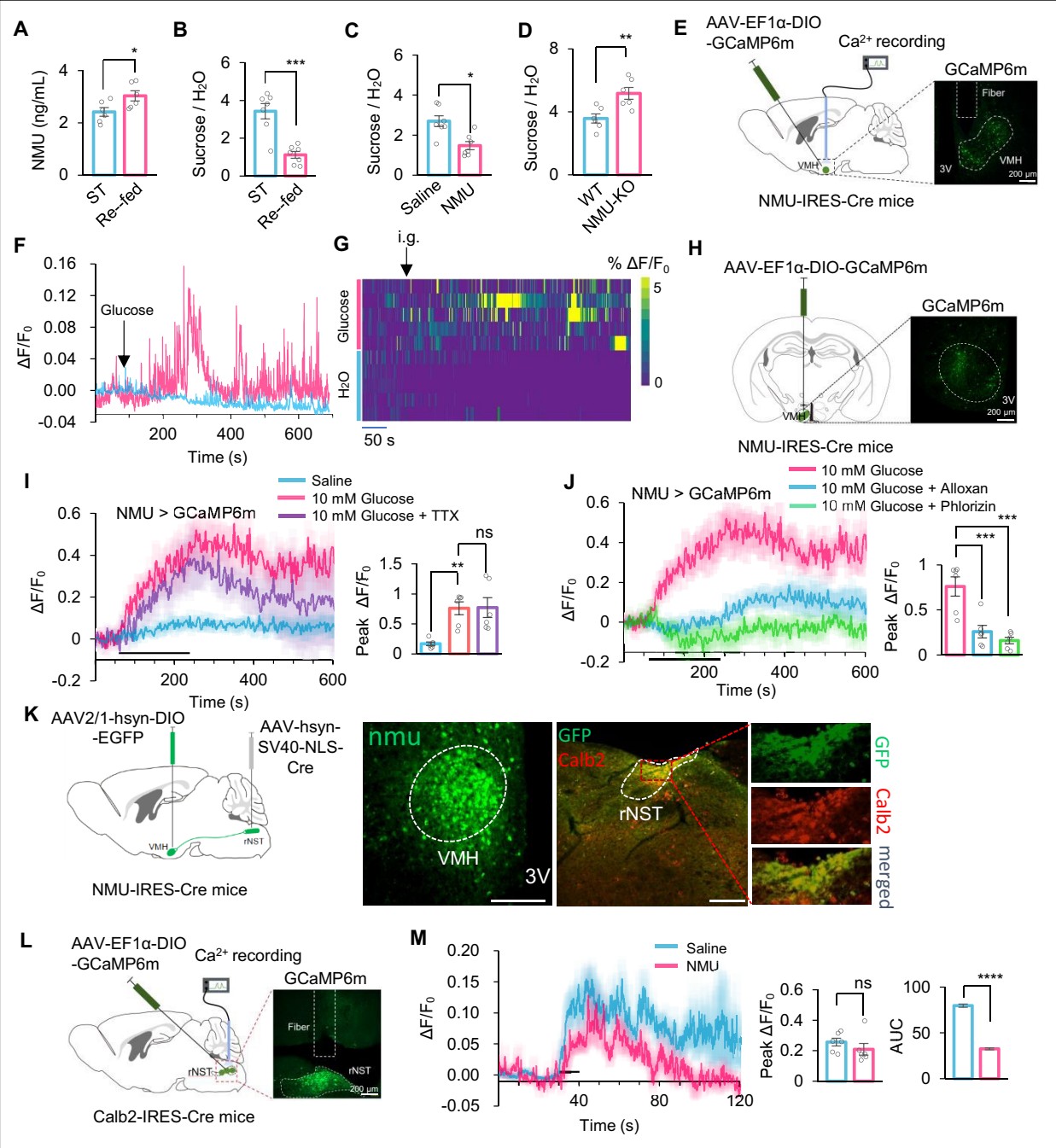

**Figure 7.** NMU+ neurons were the central energy sensor in mouse. (**A**) Blood Neuromedin U (NMU) levels of starved mice refed with 20% sucrose (*t*-test; *, p=0.0373; n=6). (**B**) Two-bottle preference tests for starved mice refed with sucrose (*t*-test; ***, p=0.0002; n=7). (**C**) Two-bottle preference tests for starved mice with or without intraperitoneal injection of NMU peptide (YFLFRPRN-NH$_2$, 4.5 µm/kg) (*t*-test; *, p=0.0293; n=7). (**D**) Two-bottle preference tests for indicated starved mice (*t*-test; **, p=0.0073; n=6). (**E**) Fiber photometry to record the calcium dynamics of NMU+ neurons in the ventromedial hypothalamus (VMH) with GCaMP6m (left) and representative IHC image of expressing GCaMP6m in VMH (right). (**F–G**) Representative trace (left) and heatmaps (right, n=5) showing calcium dynamics of NMU+ neurons in the VMH in response to gastric glucose infusion (20% sucrose, 200 µL), which elevates circulating glucose levels independently of oral sensory stimulation. (**H**) Experimental approach to assess calcium signaling in NMU+ neurons in the VMH in vitro (left) and representative IHC image of expressing GCaMP6m in VMH (right). (**I–J**) Representative traces and quantification of ex vivo calcium responses of NMU+ neurons during the perfusion of glucose with or without TTX (**I**), alloxan (**J**), and phlorizin (**J**) (one-way ANOVA; **, p=0.0049; ***, p=0.0001 for glucose+phlorizin and p=0.0006 for glucose+alloxan; n=6–7). Horizontal black bar represents the duration of indicated glucose solution stimulation. (**K**) Anterograde trans-synaptic tracing of downstream targets of NMU+ neurons. NMU+ neurons were labeled by GFP expression following injection of a Cre-dependent AAV2/1-DIO-GFP into NMU-Cre mice. This virus undergoes anterograde trans-synaptic transfer to postsynaptic neurons. To enable GFP expression specifically in downstream target regions, an AAV-Cre virus was locally injected into the rNST. As

*Figure 7 continued on next page*

*Figure 7 continued*

a result, postsynaptic neurons in the rNST receiving input from NMU$^+$ neurons were labeled by GFP delivered anterogradely from upstream NMU$^+$ neurons. Representative images show that GFP-labeled downstream neurons in the rNST colocalize with Calb2 immunoreactivity, indicating that NMU$^+$ neurons preferentially target Calb2$^+$ rNST neurons. (**L**) Fiber photometry to record the calcium dynamics of Calb2$^+$ neurons in the rNST with GCaMP6m (left) and representative IHC image of expressing GCaMP6m in rNST (right). (**M**) Representative traces and quantification of calcium responses of Calb2$^+$ neurons during glucose licking under physiological feeding conditions (500 mM sucrose), with or without NMU administration, assessing downstream modulation of sweet-responsive brainstem circuits (*t*-test; ****, $p<0.0001$; n=6). Student's *t*-test and one-way ANOVA followed by post hoc test with Bonferroni correction were used for multiple comparisons when applicable.

The online version of this article includes the following source data and figure supplement(s) for figure 7:

**Source data 1.** Source data contain numerical values and statistical results for *Figure 7*.

**Figure supplement 1.** Neuromedin U (NMU) peptide suppressed sweet taste in fly.

**Figure supplement 1—source data 1.** Source data contain numerical values and statistical results for *Figure 7—figure supplement 1*.

**Figure supplement 2.** The expressing pattern of Neuromedin U (NMU) neurons and Calb2 neurons.

**Figure supplement 3.** A working model.

A notable feature of this circuit is that its manipulation produces significant but relatively modest changes in sweet sensitivity compared with the dramatic enhancement observed under starvation (*Inagaki et al., 2014*; *Kain and Dahanukar, 2015*). This difference reflects the distinction between a specific satiety mechanism and the global physiological state of hunger. Starvation is a systemic crisis that engages multiple hunger-promoting pathways, including dopaminergic and NPF signaling, which broadly elevate motivational drive and sensory gain (*Inagaki et al., 2012*; *Williams et al., 2020*). In contrast, the hugin-AstA pathway functions as a glucose-dependent inhibitory branch whose endogenous activity is elevated in fed states but remains functionally capable of modulating sweet sensitivity across feeding conditions. Disrupting this inhibitory pathway prevents full suppression of sweet sensitivity after feeding but does not recreate the potent hunger drive generated by dopaminergic activation. Thus, energy homeostasis arises from the coordinated action between hunger accelerators and satiety brakes.

This framework reconciles our findings with previous reports that silencing hugin neurons does not alter proboscis extension in starved flies (*Marella et al., 2012*). Although endogenous hugin neuronal activity is reduced during starvation, our data show that further manipulation of this pathway can still bidirectionally influence feeding behavior under starved conditions. This indicates that the hugin-AstA circuitry is not exclusively restricted to fed states, but rather represents a state-modulated inhibitory system whose baseline tone shifts with circulating glucose levels. Under starvation, reduced endogenous activity may partially relieve this inhibitory influence; however, the circuit retains functional capacity to modulate sweet sensitivity when experimentally perturbed. In contrast, preventing the starvation-induced reduction of hugin activity—such as by thermogenetic activation—significantly suppresses hunger-enhanced sweet sensitivity. These results indicate that the dynamic downregulation of hugin neuron activity is a critical component of the state-dependent modulation of taste. Therefore, rather than functioning as a strictly 'satiety-specific brake,' the hugin-AstA axis is better conceptualized as a glucose-responsive inhibitory module that biases sweet perception according to metabolic state. Its activity increases following feeding to dampen sensory gain, while decreased activity during starvation permits heightened sweet sensitivity. Such graded modulation enables flexible integration of internal energy status with sensory processing.

Our data further reveal functional heterogeneity within peptidergic populations. Although hugin and AstA neurons are broadly distributed in the central nervous system (including the SEZ and VNC), only specific subpopulations respond to elevated glucose (*Oh et al., 2021*; *Chen et al., 2016*). While descending inputs from the VNC may potentiate signaling, our data suggest that the glucose-responsive hugin neurons in the SEZ constitute a dedicated energy-sensing module. This heterogeneity likely explains both the partial behavioral effects of global manipulations and the ability of hugin neurons to regulate diverse behaviors beyond feeding, such as locomotion.

Circulating sugars represent a complex internal signal. In *Drosophila*, trehalose serves as the major circulating sugar supporting stress resistance, whereas glucose functions as the primary, rapidly mobilized energy substrate (*Matsushita and Nishimura, 2020*). Unlike receptor-based sensors (e.g. Gr43a for fructose) (*Ma et al., 2024*), Hugin neurons rely on intracellular glucose metabolism and K$^{ATP}$

channel activity to modulate neuronal excitability. This mechanism ensures that the circuit activity is tightly coupled to cellular energy status. Consistent with this, feeding with fructose, trehalose, or non-nutritive sweeteners only weakly suppressed sweet sensitivity within the acute post-feeding interval, whereas sucrose—which rapidly restores hemolymph glucose—robustly activated this pathway. Thus, our work identifies glucose as the key metabolic signal that engages this central brake.

The presence of multiple sugar-sensing systems in *Drosophila* reflects functional specialization rather than redundancy. Peripheral and gut-derived signals regulate feeding over longer timescales, while central hunger circuits adjust sensory gain during deprivation. The hugin-AstA circuit adds a distinct layer by acting as a rapid central brake that prevents overconsumption when energy stores are sufficient.

Strikingly, we find that this circuit logic is conserved in mammals. Although NMU has been implicated in feeding suppression (*Hanada et al., 2004*; *Howard et al., 2000*), whether NMU neurons sense internal nutrients directly had remained unclear. Our data demonstrate that VMH NMU+ neurons are activated by elevations in internal glucose and form a functional circuit with sweet-responsive Calb2 neurons in the rNST. Gastric glucose infusion experiments established that NMU neurons respond to internal metabolic signals independently of oral sensory input, while physiological sugar ingestion confirmed that this pathway actively suppresses downstream sweet-driven activity. Together, these findings define a previously unrecognized NMU-centered circuit linking internal glucose sensing to the modulation of early taste processing. All mouse experiments in this study were performed in male animals; whether sex-specific differences exist in NMU-mediated glucose sensing and taste modulation remains an important question for future investigation.

The conservation of this 'satiety brake' suggests that suppressing sensory gain via central neuropeptidergic energy sensors is a fundamental strategy for maintaining energy balance. By defining a direct link between internal glucose levels and sweet taste perception, our work provides a mechanistic framework for understanding how metabolic state shapes sensory experience and feeding behavior.

# Materials and methods

**Key resources table**

| Reagent type (species) or resource | Designation | Source or reference | Identifiers | Additional information |
|---|---|---|---|---|
| Genetic reagent (*Drosophila melanogaster*) | TH-KO | Other | | Gift from Yi Rao |
| Genetic reagent (*D. melanogaster*) | PK2-R1-KO | Other | | Gift from Yi Rao |
| Genetic reagent (*D. melanogaster*) | AstA-R1-KO | Other | | Gift from Yi Rao |
| Genetic reagent (*D. melanogaster*) | UAS-PK2-R1-RNAi | Bloomington Drosophila Stock Center | Cat: #29624 | |
| Genetic reagent (*D. melanogaster*) | UAS-PK2-R2-RNAi | Bloomington Drosophila Stock Center | Cat: #28781 | |
| Genetic reagent (*D. melanogaster*) | UAS-AstA-R1-RNAi | Bloomington Drosophila Stock Center | Cat: #27280 | |
| Genetic reagent (*D. melanogaster*) | UAS-Glut1-RNAi | TsingHua Fly Center | Cat: #3043 | |
| Genetic reagent (*D. melanogaster*) | UAS-Hex-C-RNAi | TsingHua Fly Center | Cat: #3684.N | |
| Genetic reagent (*D. melanogaster*) | AstA-Gal4 | Other | | Gift from Yi Rao |
| Genetic reagent (*D. melanogaster*) | LK-Gal4 | Other | | Gift from Yi Rao |
| Genetic reagent (*D. melanogaster*) | Hugin-Gal4 | Other | | Gift from Yi Rao |

*Continued on next page*

*Continued*

| Reagent type (species) or resource | Designation | Source or reference | Identifiers | Additional information |
|---|---|---|---|---|
| Genetic reagent (*D. melanogaster*) | TK-Gal4 | Other | | Gift from Yi Rao |
| Genetic reagent (*D. melanogaster*) | CCHa1-Gal4 | Other | | Gift from Yi Rao |
| Genetic reagent (*D. melanogaster*) | CCHa2-Gal4 | Other | | Gift from Yi Rao |
| Genetic reagent (*D. melanogaster*) | PK2-R1-Gal4 | Other | | Gift from Yi Rao |
| Genetic reagent (*D. melanogaster*) | TH-Gal4 | Other | | Gift from Yi Rao |
| Genetic reagent (*D. melanogaster*) | Elav-Gal4 | Bloomington Drosophila Stock Center | Cat: #25750 | |
| Genetic reagent (*D. melanogaster*) | Dilp2-Gal4 | Bloomington Drosophila Stock Center | Cat: #37516 | |
| Genetic reagent (*D. melanogaster*) | Gr5a-Gal4 | Bloomington Drosophila Stock Center | Cat: #57591 | |
| Genetic reagent (*D. melanogaster*) | Gr5a-LexA | Bloomington Drosophila Stock Center | Cat: #93014 | |
| Genetic reagent (*D. melanogaster*) | AstA-LexA | Other | | Gift from Yi Rao |
| Genetic reagent (*D. melanogaster*) | AstA-R1-LexA | Other | | Gift from Yi Rao |
| Genetic reagent (*D. melanogaster*) | UAS-TRPA1 | Other | | Gift from David Anderson |
| Genetic reagent (*D. melanogaster*) | UAS-shibire$^{ts}$ | Other | | Gift from David Anderson |
| Genetic reagent (*D. melanogaster*) | UAS-GCaMP6m | Bloomington Drosophila Stock Center | Cat: #42748 | |
| Genetic reagent (*D. melanogaster*) | UAS-CD8-GFP, LexAop-CD2-RFP | Bloomington Drosophila Stock Center | Cat: #67093 | |
| Genetic reagent (*D. melanogaster*) | UAS-CaMPARI | Bloomington Drosophila Stock Center | Cat: #58761 | |
| Genetic reagent (*D. melanogaster*) | LexAop-Chrimson | Other | | Gift from Wei Zhang |
| Genetic reagent (*D. melanogaster*) | UAS-GCaMP6m; LexAop-CsChrimson | Other | | Gift from Yufeng Pan |
| Genetic reagent (*Mus musculus*) | NMU-KO | Cyagen | Cat: #S-KO-10720 | |
| Genetic reagent (*M. musculus*) | NMU-Cre | Shanghai Model Organisms Center | Cat: #NM-KI-200298 | |
| Genetic reagent (*M. musculus*) | Calb2-Cre | Shanghai Model Organisms Center | Cat: #NM-KI-200102 | |
| Antibody | Anti-Bruchpilot (Mouse monoclonal) | DSHB | Cat: # nc82 RRID:AB_2314866 | IF(1:100) (5 µL) |
| Antibody | Anti-GFP (Rabbit polyclonal) | Abcam | Cat: # G6556 RRID:AB_305564 | IF(1:500) (1 µL) |
| Antibody | Anti-dsRed (Rabbit polyclonal) | Clontech | Cat: #632496 RRID:AB_10013483 | IF(1:500) (1 µL) |

*Continued on next page*

*Continued*

| Reagent type (species) or resource | Designation | Source or reference | Identifiers | Additional information |
|---|---|---|---|---|
| Antibody | Anti-Calb2 (Rabbit polyclonal) | ImmunoStar | Cat: # 24445 RRID:AB_572223 | IF(1:500) (5 μL) |
| Antibody | Anti-mouse Alexa Fluor 488 | Thermo Fisher Scientific | Cat: #A11001 RRID:AB_2534069 | IF(1:500) (1 μL) |
| Antibody | Anti-rabbit Alexa Fluor 546 | Thermo Fisher Scientific | Cat: #A11010 RRID:AB_2534077 | IF(1:500) (1 μL) |
| Antibody | Anti-rabbit Alexa Fluor 488 | Thermo Fisher Scientific | Cat: #A11008 RRID:AB_143165 | IF(1:500) (1 μL) |
| Antibody | Anti-mouse Alexa Fluor 647 | Thermo Fisher Scientific | Cat: #A21235 RRID:AB_2535804 | IF(1:500) (1 μL) |
| Other | AAV2/9-EF1α-DIO-GCaMP6m-WPRE | Obio Biotechnology | Cat: #H4955 | Bacterial and virus strains |
| Other | AAV2/1-hsyn-EGFP-2A-Cre-WPRE | Obio Biotechnology | Cat: #H4942 | Bacterial and virus strains |
| Other | AAV2/1-hsyn-SV40 NLS-Cre | Brain Case | Cat: #BC-0159 | Bacterial and virus strains |
| Other | AAV2/9-EF1α-DIO-mcherry-WPRE | Brain Case | Cat: #BC-0016 | Bacterial and virus strains |
| Other | AAV2/9-EF1α-DIO-EGFP-WPRE | Brain Case | Cat: #BC-0015 | Bacterial and virus strains |
| Chemical compound | Brilliant Blue | MACKLIN | Cat: #E808678 | |
| Chemical compound | Phlorizin | TargetMol | Cat: #T2922 | |
| Chemical compound | Alloxan | TargetMol | Cat: #T7814L | |
| Chemical compound | Glibenclamide | TargetMol | Cat: #T1634 | |
| Chemical compound | All-*trans*-retinal | Sigma-Aldrich | Cat: #R2500 | |
| Chemical compound | L-Glucose | Sigma-Aldrich | Cat: #G5500 | |
| Chemical compound | Pyruvate | Sigma-Aldrich | Cat: #P5280 | |
| Chemical compound | Triton X-100 | Sigma-Aldrich | Cat: #T8787 | |
| Chemical compound | Calf serum | Thermo Fisher Scientific | Cat: #16010159 | |
| Commercial assays or kit | Glucose Content Assay Kit | Solarbio | Cat: #BC2500 | |
| Commercial assays or kit | NMU ELISA Assay Kit | BMASSY | Cat: #74030 | |
| Commercial assays or kit | NMU RNAscope | Pinpoease | Cat: #PIF2000 | |
| Software, algorithm | Fiji/ImageJ | NIH | | |
| Software, algorithm | GraphPad Prism 9 | GraphPad Software | | |

## Animals

Fly: Flies were maintained on a standard fly diet under conditions of 60% humidity and 25°C with a 12 hr light and 12 hr dark cycle. Virgin female flies were chosen immediately after emerging and housed in standard fly food (ND) vials, with 20 flies per vial, for a period of 5–6 days before experiments. For temperature-sensitive manipulations (dTrpA1, Shibire^ts), flies were reared at 18°C for 7–8 days, then transferred to 30°C for 30 min to activate or inhibit targeted neurons; behavioral assays (PER or feeding) were conducted continuously at 30°C. For starvation, the flies were kept on 2% agar for 12 hr. For refeeding, the fasted flies were transferred to different food for 15 min.

The following UAS-RNAi lines: *UAS-AstA-R1 RNAi* (#27280), *UAS-AstA-R2 RNAi* (#25935), *UAS-PK2-R1 RNAi* (#29624), *UAS-PK2-R2 RNAi* (#28781), *UAS-CD8-GFP, LexAop-CD2-RFP* (#67093), *UAS-GCaMP6m* (#42748), *UAS-CaMPARI* (#58764), *Gr5a-GAL4* (#57592), *Gr5a-LexA* (#93014), *ilp2-GAL4* (#37516), *elav-GAL4* (#25750) were obtained from the Bloomington Drosophila Stock Center at

Indiana University. *TH-KO, PK2-R1-KO, AstA-R1-KO,* and all the other Gal4 and LexA lines (*AstA-Gal4, LK-Gal4, hugin-Gal4, TK-Gal4, CCHa1-Gal4, CCHa2-Gal4, TH-Gal4, PK2-R1-Gal4, AstA-LexA, AstA-R1-LexA*) were from Yi Rao (*Deng et al., 2019*) (Capital Medical University, Beijing). *UAS-dTrpA1* and *UAS-Shibire^{ts1}* were from David Anderson (Caltech). *UAS-GCaMP6m; LexAop-CsChrimson* was from Yufeng Pan (Southeast University, Nanjing). *LexAop-Chrimson* was from Wei Zhang (Tsinghua University, Beijing).

Mouse: Male C57BL/6J mice and NMU-KO mice at 4 weeks of age were procured from Cyagen. The NMU-Cre and Calb2-Cre mice were procured from Shanghai Model Organisms Center. The mice were kept in standard laboratory conditions at a temperature of 25°C with a 12:12 light/dark cycle. All experimental procedures were conducted in compliance with the guidelines set by the Laboratory Animal Welfare and Ethics Committee of Chongqing University (CQU-IACUC-RE-202401-004), adhering to both national and international standards.

## Behavioral assays (Fly)

PER was evaluated using a method outlined in a previous study (*Marella et al., 2012*). Briefly, individual flies were gently aspirated and positioned in a 200 μL pipette tip, allowing only the head and proboscis to protrude while the dorsal thorax was lightly affixed to prevent escape. After a 3 min acclimation period, each fly was first presented with 0.5 μL water droplets on the labellum twice. Flies that responded by extending their proboscis were allowed to drink and were subsequently excluded from the experiment. Flies that did not extend their proboscis in response to water were then stimulated sequentially with ascending sucrose concentrations (6.25–800 mM). Each concentration was delivered as two brief (<1 s) touches of a 0.5 μL droplet to the labellum, immediately withdrawn to prevent ingestion. A full proboscis extension, defined as the complete uncoiling of the rostrum beyond the labellum plane, was scored as positive. Partial or delayed (>1 s) extensions were not considered positive responses. Responsiveness at each concentration was recorded if at least one full extension occurred in the two trials. For *Figure 4—figure supplement 1*, the connection between the brain and VNC was cut off by dissecting scissors before the PER assay.

Feeding was assessed as previously described (*Sun et al., 2017*). Briefly, 5-day-old flies underwent a 12 hr starvation period on 2% agar and were then moved to a new vial with test food containing 0.5% Brilliant Blue (MACKLIN, China) for 10 min. After a rapid freeze at –20°C, flies were decapitated in PBS, homogenized, and centrifuged (13,000×*g* * 5 min). The resulting supernatants were diluted with PBS to a total volume of 1 mL, and absorbance was measured at 620 nm.

## Sucrose preference test

The two-bottle preference (TBP) behavioral test was conducted in a standard mouse cage. Two bottles, one containing water and the other with a 10 mM sucrose solution, were placed on top of the cage. Each tested mouse was individually caged and adapted to two bottles containing distilled water or a 10 mM sucrose solution for 1 week. Throughout the training period, the positions of the bottles were alternated every 24 hr to mitigate potential position-related biases. Prior to the TBP tests for sucrose, mice underwent a 12 hr pre-starvation period, followed by refed sucrose (20%) for 1 hr or intraperitoneal (IP) administration of NMU (YFLFRPRN-NH$_2$, 4.5 μM/kg). The consumption of water and sucrose solution over a 2 hr period was meticulously recorded. The sweetener preference ratio was then calculated by dividing the weight of sweetener consumed by the weight of water consumed.

## NMU measurement

Blood was obtained from the lower jaw of the mice and clotted at room temperature for 1 hr. The samples were then centrifuged (1500×*g*, 4°C) for 10 min, and the supernatants were collected for NMU measurement. All the manipulations were according to the manufacturer's instructions (BMASSY, 74030).

## Hemolymph extraction and glucose measurement

Groups of 40 ice-anesthetized flies were decapitated and placed into a perforated 0.5 mL tube, which was nested inside a 1.5 mL collection tube. Samples were centrifuged at 2500×*g* for 10 min at 4°C, yielding approximately 2 μL of hemolymph per batch. Hemolymph was pooled to 4 μL per assay

(n=8 independent samples per group) and immediately analyzed for glucose concentration using the Solarbio BC2500 kit (Solarbio, China) according to the manufacturer's instructions.

## Microinjection

Flies were delicately positioned and secured within a 200 mL pipette tip, ensuring their heads were directed toward the tip's end. Subsequently, the tip was opened by cutting, exposing the heads and a section of the thorax. About 20 nL of sterile saline, either with or without synthesized peptides (hugin: SVPFKPRL-NH$_2$, 1 mg/mL [1.1 mM]; AstA: LPVYNFGL-NH$_2$, 1 mg/mL [1.1 mM]; NMU: YFLFRPRN-NH$_2$, 1 mg/mL [0.9 mM]; control peptide: DYKDDDDKYPYDVPDYA, 1 mg/mL [0.48 mM]), were injected into the thorax of these flies using a glass micropipette and a microinjector (3-000-207, Drummond Scientific Company Instruments). The glass micropipette was created from thick-walled borosilicate capillaries (3-000-203-G/X, Drummond Scientific Company Instruments).

## Immunofluorescence staining

Fly brains were carefully dissected in PBS on ice and then fixed in 4% formaldehyde for 60 min. Following fixation, the brains underwent permeabilization and blocking using Dilution/Blocking Buffer (PBS containing 10% Calf Serum and 0.5% Triton X-100) for 2 hr at room temperature. Subsequently, the samples were immersed in the appropriate primary antibodies in Dilution/Blocking Buffer for 24 hr at 4°C. Afterward, the samples were subjected to a 60 min wash with Washing Buffer (0.5% Triton X-100 in PBS) four times at room temperature and were then incubated with secondary antibodies for 24 hr at 4°C. The samples underwent three additional washes with Washing Buffer before being mounted in Fluoroshield (Sigma-Aldrich).

Images were acquired using a scanning confocal microscope with Olympus objectives (20× /0.7 and 40× /0.95w). Antibodies were employed at the following dilutions: mouse anti-nc82 (1:100, DSHB), rabbit anti-GFP (1:500, Abcam), Alexa Fluor 647 goat anti-mouse (1:500, Invitrogen), and Alexa Fluor 488 goat anti-rabbit (1:500, Invitrogen).

## In situ hybridization

Mice were anesthetized and perfused with 4% paraformaldehyde (PFA) dissolved in phosphate-buffered saline (PBS, pH 7.4). Brains were carefully removed from the skull and post-fixed in 4% PFA at 4°C overnight. Subsequently, the brains were transferred to a 20% sucrose solution (in 1× PBS) at 4°C until they were ready for sectioning. Twenty-micrometer-thick sections were obtained using a cryostat. The detection of NMU was performed using a fluorescent in situ hybridization technique (RNAscope, Pinpoease) according to the manufacturer's instructions. Sections were mounted on SuperFrost Plus Gold slides (Thermo Fisher) and briefly rinsed in autoclaved Millipore water. They were then subjected to gradient dehydration in 50%, 75%, and 100% ethanol for 5 min each. A hydrophobic barrier was created around the sections using an ImmEdge hydrophobic barrier pen (Cat No. 310018). All incubation steps were carried out at 40°C using the Pinpoease hybridization system (Cat No. SH-08). The subsequent hybridization, amplification, and detection steps were performed strictly according to the manufacturer's instructions.

## Stereotaxic brain surgeries and viral injection

Mice were anesthetized with 1–2% isoflurane during stereotaxic injections, performed using a small animal stereotaxic instrument (RWD Life Science, #68030). Throughout the surgery, core body temperature was maintained at 36 ± 1°C using a feedback-controlled heating system. Micro scissors were used for the incision, and a dental drill was employed to create the cranial window. Viral injections were delivered using a glass microelectrode syringe pump (RWD Life Science, #R-480). To express GCaMP6m and mCherry AAV2/9-EF1α-DIO-GCaMP6m-WPRE (titer: 2.89×10$^{12}$, #H4955, Obio Biotechnology, Shanghai, China, Co., Ltd.) and AAV2/9-EF1α-DIO-mCherry-WPRE (titer: 2.89×10$^{12}$, Obio Biotechnology, Shanghai, China, Co., Ltd.) were bilaterally injected into the VMH and rNST, with approximately 0.15 µL of virus delivered per site. The microelectrode was left in place for 10 min post-injection before being withdrawn. Injection coordinates were as follows: VMH (AP, −1.46 mm; ML,±0.4 mm; DV, −5.5 mm) and rNST (AP, −7.08 mm; ML, ±0.6 mm; DV, −4.3 mm). After 3 weeks of AAV injections, optical fibers were chronically implanted in the VMH or rNST and secured with dental cement (C&B Metabond, Parkell, Japan). Injection coordinates were determined based

on the 4th edition of the mouse brain atlas by Franklin and Paxinos (2008). Mice were allowed to recover for 5–7 days post fiber-implantation before any behavioral or calcium imaging evaluations were conducted. After the experiments, mice were perfused to verify virus expression and fiber placement. Data from mice with poor viral expression or incorrect fiber placement (0–20% of cases) were excluded from the final analysis.

For the anterograde tracing experiments, two viral vectors were used. AAV2/1-hsyn-DIO-EGFP-WPRE (titer: $1 \times 10^{13}$ viral particles/mL, Obio Biotechnology) was injected into the VMH, and AAV2/1-hsyn-SV40 NLS-Cre (titer: $2.1 \times 10^{12}$ viral particles/mL, Brain Case) was injected into the rNST. Approximately 0.15 µL of virus was delivered per injection site. After 3 weeks, the brains were perfused and fixed. Subsequently, the brain tissues were sliced and subjected to antibody staining. The fluorescence signals were then recorded and analyzed.

## Optical-fiber-based calcium recording in freely moving mice

Following injection of an AAV2/9-EF1α-DIO-GCaMP6m-WPRE viral vector, an optical fiber (200 µm OD, 0.37 numerical aperture, Inper Inc, China) was placed 150 µm above the viral injection site. GCaMP6m$^+$ mice were implanted with the optic fiber 3 weeks post-AAV injection and allowed to recover for 5–7 days prior to experimental testing.

For *Figure 7K and L*, six mice were first trained to lick glucose (500 mM) from a Petri dish. After training, they were fasted for 12 hr and then received an IP injection of synthetic NMU peptide (4.5 µm/kg), followed by a 30 min waiting period. During the experimental session, the mice were given 10 s to lick glucose from the Petri dish while undergoing fiber photometry recording. For *Figure 5E–G*, mice were fasted for 4 hr and then administered glucose via gastric infusion, with fiber photometry recordings capturing the process. Fluorescence signals were acquired using a dual-channel fiber photometry system (410 nm and 470 nm, RWD Life Science). The laser power at the tip of the optical fiber was kept below 20 µW for both the 470 nm and 410 nm channels to minimize bleaching. ΔF/F was calculated as (470 nm signal – fitted 410 nm signal)/fitted 410 nm signal. Data analysis was performed using software from RWD Life Science.

## Ex vivo calcium imaging

For *Figure 1G, I*, *Figure 1—figure supplement 6*, and *Figure 2*, freshly dissected fly brains were placed in the sugar-free AHL buffer (108 mM NaCl, 8.2 mM MgCl$_2$, 4 mM NaHCO$_3$, 1 mM NaH$_2$PO$_4$, 5 mM KCl, 2 mM CaCl$_2$, 5 mM HEPES, pH 7.3). Baseline recordings of the samples in AHL buffer were taken over 1 min. Subsequently, the solutions were added with 100 µM glibenclamide (*Figure 2D*), 50 mM pyruvate (*Figure 2G*), or switched to AHL+glucose (50 mM) with or without indicated chemicals (*Figures 2A and E*, 1 mM phlorizin or 10 µM alloxan), and the pH was adjusted back to 7.3 through gentle perfusion for 3 min. Solution flow in the perfusion chamber was controlled by a valve commander (Scientific Instruments). Following stimulation, samples were washed out again with AHL. For the TTX test, 2 µM TTX was added to the AHL solution. Prior to the assay, the samples were pretreated with 2 µM TTX for 15 min.

For *Figure 7H–J*, mice (6–9 weeks of age) were deeply anesthetized by avertin (1.25% avertin, 0.2 mL/10 g). Mouse brains were quickly extracted following decapitation and immediately placed in ice-cold slicing buffer (110 mM choline Cl, 2.5 mM KCl, 1 mM NaH$_2$PO$_4$, 25 mM NaHCO$_3$, 5 mM glucose, 7 mM MgCl$_2$, 0.5 mM CaCl$_2$, 1.3 mM Na ascorbate, 0.6 mM Na pyruvate bubbled with 95% oxygen, and 5% CO$_2$ with an adjusted pH of 7.3). The brains were then sectioned into 200 µm slices using a vibratome (Leica VT1200S). These slices were incubated in artificial cerebrospinal fluid (aCSF) (125 mM NaCl, 2.5 mM KCl, 1 mM NaH$_2$PO$_4$, 25 mM NaHCO$_3$, 5 mM glucose, 1.3 mM MgCl$_2$, 2 mM CaCl$_2$, 1.3 mM Na ascorbate, 0.6 mM Na pyruvate bubbled with 95% oxygen and 5% CO$_2$ with an adjusted pH of 7.3) at 34°C for 20 min, then transferred to room temperature for 30 min before recording. For Ca$^{2+}$ imaging, the slices were placed in a recording chamber in the low sugar aCSF (125 mM NaCl, 2.5 mM KCl, 1 mM NaH$_2$PO$_4$, 25 mM NaHCO$_3$, 1 mM glucose, 1.3 mM MgCl$_2$, 2 mM CaCl$_2$, 1.3 mM Na ascorbate, 0.6 mM Na pyruvate, pH 7.3) for 10 min (maintained the flow rate for 1–3 mL/min). Baseline recordings, perfusion, and Ca$^{2+}$ recording were performed as detailed above.

All imaging was conducted on an Olympus confocal microscope (FVMPE-RS) with a water immersion objective lens (25×/1.05w MP). Image analyses were performed using ImageJ to calculate the mean intensity of the indicated neuron targeting ROIs and then plotted in Excel (Microsoft). Ratio

changes were calculated using the formula: $\Delta F/F = [F - F_0]/F_0$, where F represents the mean fluorescence of the cell body, and $F_0$ is the average baseline (before stimulation).

## Optogenetics and in vivo calcium imaging

Newly emerged virgin female flies were housed in a fresh vial with standard medium for 3 days and subsequently moved to a vial containing regular food supplemented with 400 µM all-*trans*-retinal (Sigma R2500) for 2–4 days prior to experiments. These flies were then immobilized on ice, affixed to transparent tape, and the dorsal cuticle of the fly head was delicately removed with forceps to expose the brain, which was immersed in AHL.

For *Figure 5G and H*, to apply liquid food (5% sucrose) to the proboscis, a pipette was filled with the taste solution and placed a few microns from the proboscis tip by using a micromanipulator (MP225, Sutter Instruments) prior to recording. Then, we recorded calcium signals for 60 s in the absence of light to establish a baseline (1 s per frame). To activate neurons expressing CsChrimson, a red LED was positioned above the brain, controlled by a custom computer program during opto-activation (0.45 s on and 0.55 s off from 60 s to 120 s). The pipette was placed on the proboscis from 80 s to 85 s. Calcium signals of Gr5a$^+$ neurons were recorded throughout these processes (1 s per frame).

Image analyses about relative fluorescence changes $\Delta F/F$ were performed as ex vivo calcium imaging.

## Optogenetics and ex vivo calcium imaging

For *Figure 4—figure supplement 1C*, newly emerged virgin female flies were housed in a fresh vial with standard medium for 3 days and subsequently moved to a vial containing regular food supplemented with 400 µM all-*trans*-retinal (Sigma R2500) for 2–4 days prior to experiments. The brains were dissected and placed under a red LED. Calcium signals of AstA$^+$ neurons were recorded with or without LED light as above. Image analyses were performed as ex vivo calcium imaging.

## Calcium imaging with CaMPARI

The flies were then gently immobilized on ice, attached to transparent tape, and the dorsal cuticle of the fly head was carefully removed with forceps to expose the brain, which was immersed in AHL. PC was achieved for 30 s using a 405 nm LED (Thorlabs, M405L2-UV [405 nm] Mounted LED, 1000 mA, 410 mW), controlled by a LED controller (Thorlabs, LEDDB1 driven with 1000 mA). Images were captured using an Olympus confocal microscope. Image analyses were performed in ImageJ by manually drawing ROIs covering individual neuronal cell bodies using the green channel. The extent of CaMPARI PC was determined as the red:green ratio. A '−405 nm' control was included to demonstrate that scanning of the hugin neurons without exposure to ultraviolet light does not convert green to red fluorescence.

## Statistical analysis

Data are shown as means (± SEM). Data presented in this study were verified for normal distribution by D'Agostino-Pearson omnibus test. Student's *t*-test, one-way ANOVA, and two-way ANOVA (for comparisons among three or more groups and comparisons with more than one variant) were used. The post hoc test with Bonferroni correction was performed for multiple comparisons following ANOVA.

## Acknowledgements

We express gratitude to all members of the Wang and Huang Lab for their valuable discussions and technical support. Special thanks to Yi Rao and Bowen Deng for generously providing *Drosophila* chemoconnectome lines, and to Yulong Li and Jianzhi Zeng for their assistance with in vivo calcium imaging preparations. Appreciation is extended to the members of the Neuroscience Pioneer Club for their insightful discussions throughout the course of this study. This study was funded by the National Natural Science Foundation of China (No. 32271008 for RH and No. 32071006 for LW), Guangdong Basic and Applied Basic Research Foundation (2023A1515110454 for TS), and the startup funds from Shenzhen Bay Laboratory and Chinese Institutes for Medical Research to LW.

## Additional information

### Funding

| Funder | Grant reference number | Author |
|--------|------------------------|--------|
| National Natural Science Foundation of China | 32271008 | Rui Huang |
| National Natural Science Foundation of China | 32071006 | Liming Wang |
| Guangdong Basic and Applied Basic Research Foundation | 2023A1515110454 | Tingting Song |
| Shenzhen Bay Laboratory and Chinese Institutes for Medical Research | | Liming Wang |

The funders had no role in study design, data collection and interpretation, or the decision to submit the work for publication.

### Author contributions

Wusa Qin, Resources, Data curation, Validation, Investigation, Visualization; Tingting Song, Resources, Data curation, Validation, Investigation, Writing – original draft; Zeliang Lai, Daihan Li, Data curation, Investigation; Liming Wang, Supervision, Funding acquisition, Project administration, Writing – review and editing; Rui Huang, Supervision, Funding acquisition, Writing – original draft, Project administration, Writing – review and editing

### Author ORCIDs

Liming Wang (iD) https://orcid.org/0000-0002-7256-8776

### Ethics

All animal experiments were conducted in accordance with national and international guidelines for the care and use of laboratory animals. All procedures were approved by the Laboratory Animal Welfare and Ethics Committee of Chongqing University (Approval No.: CQU-IACUC-RE-202401-004). All procedures were carried out in compliance with relevant institutional and national regulations. Every effort was made to minimize animal suffering and to reduce the number of animals used.

Reviewer #1 (Public review): https://doi.org/10.7554/eLife.108551.4.sa1
Reviewer #2 (Public review): https://doi.org/10.7554/eLife.108551.4.sa2
Author response https://doi.org/10.7554/eLife.108551.4.sa3

---

## Additional files

### Supplementary files

MDAR checklist

### Data availability

Source data contain the numerical data used to generate the figures.

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
