## [Editor Report · eLife Assessment]

This **important** work delineates layered glucose-responsive neuropeptidergic mechanisms that regulate sugar intake. Using a combination of genetic, physiological, and behavioral experiments, the authors **convincingly** show that Hugin- and Allatostatin A-releasing neurons suppress sugar feeding by reducing the sensitivity of Gr5a-expressing gustatory neurons. They further demonstrate that Neuromedin U neurons share key physiological properties with fly Hugin neurons, highlighting conserved peptide functions across animal phyla.

---

## [Referee Report · Reviewer #1 (Public review)]

This revised manuscript by Qin and colleagues delineates an important neural mechanism that suppresses the intake of sugar solution in response to internal glucose level (the "brake" mechanism for sugar consumption). They identified a three-step neuropeptidergic system that downregulates the sensitivity of sweet-sensing gustatory sensory neurons, primarily in response to elevated level of circulating glucose. First, neurons that release a neuropeptide Hugin (which is an insect homolog of vertebrate Neuromedin U (NMU)) are activated by a high concentration of hemolymph glucose, which is directly sensed by Hugin-releasing neurons in a cell-autonomous mechanism. Next, Hugin neuropeptides activate Allatostatin A (AstA)-releasing neurons via one of Hugin receptors, PK2-R1. Finally, the released AstA neuropeptide suppresses sugar response in sweet-sensing Gr5a-expressing gustatory sensory neurons through the AstA-R1 receptor. Suppression of sugar response in Gr5a-expressing neurons reduces fly's sugar intake motivation. They also found that NMU-expressing neurons in the ventromedial hypothalamus (VMH) of mice (which project to the rostal nucleus of the solitary tract (rNST)) are also activated by high concentration of circulating glucose, independent of synaptic transmission, and that injection of NMU reduces the glucose-induced activity in the downstream of NMU-expressing neurons in rNST. These data suggest that the function of Hugin neuropeptides in the fly is analogous to the function of NMU in the mouse.

The authors have provided multiple lines of compelling evidence generated through rigorous and comprehensive experiments, which spans genetic abrogation, neuronal manipulation, pharmacology, and functional imaging. The authors are also receptive to the critiques and reframed the central message, such that their conclusions are soundly supported by the presented data. Importantly, the parallel study in mice adds a unique comparative perspective that makes the paper of interest to a wide range of readers.

---

## [Referee Report · Reviewer #2 (Public review)]

Summary:

The question of how caloric and taste information interact and consolidate remains both active and highly relevant to human health and cognition. The authors of this work sought to understand how nutrient sensing of glucose modulates sweet sensation. They found that glucose intake activates hugin signaling to AstA neurons to suppress feeding, which contributes to our mechanistic understanding of nutrient sensation. They did this by leveraging the genetic tools of Drosophila to carry out nuanced experimental manipulations, and confirmed the conservation of their main mechanism in a mammalian model. This work builds on previous studies examining sugar taste and caloric sensing, enhancing the resolution of our understanding.

Strengths:

Fully discovering neural circuits that connect body state with perception remains central to understanding homeostasis and behavior. This study expands our understanding of sugar sensing, providing mechanistic evidence for a hugin/AstA circuit that is responsive to sugar intake and suppresses feeding. In addition to effectively leveraging the genetic tools of Drosophila, this study further extends their findings into a mammalian model with the discovery that NMU neural signaling is also responsive to sugar intake.

Weaknesses:

The effect of Glut1 knockdown on PER in hugin neurons is modest in both fed and starved flies, suggesting that glucose intake through Glut1 may only be part of the mechanism. The authors address this in their discussion.

---

## [Author Response]

The following is the authors’ response to the previous reviews

**Public Reviews:**

**Reviewer #1 (Public review):**
In this revised manuscript, Qin and colleagues aim to delineate a neural mechanism that is engaged specifically in the sated flies to suppress the intake of sugar solution (the "brake" mechanism for sugar consumption). They identified a three-step neuropeptidergic system that downregulates the sensitivity of sweet-sensing gustatory sensory neurons in sated flies. First, neurons that release a neuropeptide Hugin (which is an insect homolog of vertebrate Neuromedin U (NMU)) are in active state when the concentration of glucose is high. This activation depends on the cell-autonomous function of Hugin-releasing neurons that sense hemolymph glucose levels directly. Next, the Hugin neuropeptides activate Allatostatin A (AstA)-releasing neurons via one of Hugin receptors, PK2-R1. Finally, the released AstA neuropeptide suppresses sugar response in sugar-sensing Gr5a-expressing gustatory sensory neurons through AstA-R1 receptor. Suppression of sugar response in Gr5a-expressing neurons reduces fly's sugar intake motivation. They also found that NMU-expressing neurons in the ventromedial hypothalamus (VMH) of mice (which project to the rostal nucleus of the solitary tract (rNST)) are also activated by high concentrations of glucose independent of synaptic transmission, and that injection of NMU reduces the glucose-induced activity in the downstream of NMU-expressing neurons in rNST. These data suggest that the function of Hugin neuropeptide in the fly is analogous to the function of NMU in the mouse.The shift of the narrative, which focuses specifically on the hugin-AstA axis as the "brake" on the satiety signal and feeding behavior, clarified the central message of the presented work. The authors have provided multiple lines of compelling evidence generated through rigorous experiments. The parallel study in mice adds a unique comparative perspective that makes the paper interesting to a wide range of readers.While I deeply appreciate the authors' efforts to substantially restructure the manuscript, I have a few suggestions for further improvements. First, there remains room for discussion whether the "brake" function of the hugin-AstA axis is truly satiety state-dependent. The fact that neural activation (Fig. Supp. 8), peptide injection (Fig. 3A, 4A), receptor knockdown (Fig. 3C,G, 4E), and receptor mutants (Fig. Supp. 10, 12) all robustly modulate PER irrespective of the feeding status suggests that the hugin-AstA axis influences feeding behaviors both in sated and hungry flies. Additionally, their new data (Fig. Supp. 13B, C) now shows that synaptic transmission from hugin-releasing neurons is necessary for completely suppressing feeding even in sated flies. If the hugin-AstA axis engages specifically in sated (high glucose) state, disruption of this neuromodulatory system is expected to have relatively little effect in starved flies (in which the "brake" is already disengaged).

We thank the reviewer for pointing out this inconsistency. We have corrected this interpretation. Specifically:

(1) We removed statements suggesting that the circuit is fully disengaged during starvation.

(2) We now state that endogenous hugin activity is reduced during starvation, but the circuit retains modulatory capacity when experimentally perturbed.

(3) The Discussion now emphasizes that the system operates as a state-modulated inhibitory tone rather than a strictly fed-state switch.

We believe this revised framing resolves the discrepancy.

In this context, it is intriguing that the knockdown of PK2-R2 hugin receptor modestly but consistently decreases proboscis extension reflex specifically in starved flies (Fig. 3D, H). The manuscript does not discuss this interesting phenotype at all. Given the heterogeneity of hugin-releasing neurons (Fig. Supp. 7), there remains a possibility that a subset of hugin-releasing neurons and/or downstream neurons can provide a complementary (or even opposing) effect on the feeding behavior.

We agree that this is an important observation. Although the effect size is modest, it is reproducible and suggests that hugin signaling may not operate as a strictly linear pathway.

To address this:

(1) We added a paragraph in the Results acknowledging the PK2-R2-dependent phenotype.

(2) We included a discussion noting the potential functional heterogeneity of hugin neurons.

(3) The schematic model (now Figure Supplementary 17, previously Figure Supplementary 16) includes a dashed line indicating a possible parallel PK2-R2-dependent branch.

Given these intriguing yet unresolved issues, it is important to acknowledge that whether this system is "selectively engaged in fed states to dampen sweet sensation (in Discussion)" requires further functional investigations. Consistent effects of manipulation of the hugin-AstA system across multiple experimental approaches underscores the importance of this molecular circuitry axis for controlling feeding behaviors. Moderation of conclusions to accommodate alternative interpretation of data will be beneficial for field to determine the precise mechanism that controls feeding behaviors in future studies.

We fully agree with the reviewer. Our original description of the circuit as a “satiety brake” implied exclusive engagement in fed states, which is not strictly supported by the behavioral data. Although endogenous hugin activity is elevated under fed conditions (as shown by CaMPARI), experimental manipulations demonstrate that the circuit retains functional capacity to modulate feeding behavior across feeding states.

To address this concern, we have:

(1) Removed the term “satiety-specific brake” throughout the manuscript.

(2) Reframed the circuit as a glucose-responsive, state-modulated inhibitory module.

(3) Revised the Discussion to explicitly state that the hugin–AstA pathway biases sweet sensitivity according to circulating glucose levels rather than functioning as an on/off switch.

(4) Substantially revised Supplementary Figure 17 to reflect graded modulation across metabolic states rather than binary state engagement.

These changes better align our conclusions with the experimental observations.

**Reviewer #2 (Public review):**
Summary:The question of how caloric and taste information interact and consolidate remains both active and highly relevant to human health and cognition. The authors of this work sought to understand how nutrient sensing of glucose modulates sweet sensation. They found that glucose intake activates hugin signaling to AstA neurons to suppress feeding, which contributes to our mechanistic understanding of nutrient sensation. They did this by leveraging the genetic tools of Drosophila to carry out nuanced experimental manipulations, and confirmed the conservation of their main mechanism in a mammalian model. This work builds on previous studies examining sugar taste and caloric sensing, enhancing the resolution of our understanding.Strengths:Fully discovering neural circuits that connect body state with perception remains central to understanding homeostasis and behavior. This study expands our understanding of sugar sensing, providing mechanistic evidence for a hugin/AstA circuit that is responsive to sugar intake and suppresses feeding. In addition to effectively leveraging the genetic tools of Drosophila, this study further extends their findings into a mammalian model with the discovery that NMU neural signaling is also responsive to sugar intake.Weaknesses:The effect of Glut1 knockdown on PER in hugin neurons is modest in both fed and starved flies, suggesting that glucose intake through Glut1 may only be part of the mechanism.

We agree that the modest PER phenotype suggests that Glut1-mediated glucose uptake represents one component of glucose sensing in hugin neurons. We have clarified this in the Discussion and now explicitly state that additional glucose-sensing mechanisms may contribute to hugin activation.

Additionally, many of the manipulations testing the "brake" circuitry throughout the study show similar effects in both fed and starved flies. This suggests that the focus of the discussion and Supplemental Figure 16 on a satiety-specific "brake" mechanism may not be fully supported by the data.

We fully agree that the previous framing overstated state specificity.

As described above, we have:

(1) Removed “satiety-specific brake” terminology.

(2) Reframed the circuit as a glucose-responsive inhibitory module.

(3) Revised the Discussion to explicitly acknowledge modulation across feeding states.

(4) Updated the schematic model (Figure Supplementary 17, formerly Figure Supplementary 16) accordingly.

**Recommendations for the authors:**

**Reviewing Editor (Recommendations for the authors):**
Both the reviewers and I agree that the conclusion about a "satiety-dependent" brake needs to be modified to discuss the phenotypes that are also observed under starved conditions. Reviewer 1 would further like to emphasize that the authors are not required to follow through with the specific recommendations suggested by them. Modifying the conclusion and Supplementary Figure 16 should suffice.

We sincerely thank the Reviewing Editor for the clear guidance. We fully agree that our previous framing of the hugin–AstA circuit as a strictly “satiety-dependent” brake may have overstated the state specificity of the system.

In response to this recommendation, we have:

(1) Revised the Abstract, Results, and Discussion to moderate the conclusion and explicitly acknowledge the phenotypes observed under starved conditions.

(2) Reframed the circuit as a glucose-responsive, state-modulated inhibitory module, rather than a satiety-exclusive brake.

(3) Supplementary Figure 17 (formerly Figure Supplementary 16) has been substantially revised to illustrate graded modulation across metabolic states rather than binary engagement.

We appreciate the clarification that no additional experiments were required and are grateful for the opportunity to improve the conceptual framing of our work.

Please include full statistical reporting in the main manuscript (e.g., figure legends or results).

We have revised all figure legends to include full statistical reporting.

**Reviewer #1 (Recommendations for the authors):**
By re-framing their finding as the "brake" mechanism on satiety-induced suppression of feeding behavior and sensitivity to sweet taste, the authors substantially improved the clarity of their findings and their significance. The additional data (Fig. Supp. 13B, C) allows "apple-to-apple" comparisons of behavioral data. I support the publication of this manuscript with no further experiments, although I have several suggestions for the text.As I write in the public review, I have a reservation on the authors' argument that hugin-AstA system is the "'satiety brake' - that is selectively engaged in fed states to dampen sweet sensation (lines 392-394)". Manipulation of both hugin system (Fig. 2C, Fig. 3A, C, D, G, Fig. Supp. 8A, C, Fig. Supp. 10A-C, Fig. Supp. 13B, C) and AstA system (Fig. 4A, E, Fig. Supp., 8C, D, Fig. Supp. 12A-C, Fig. Supp. 13D) all indicate that hugin-AstA system suppresses feeding regardless of the satiety state. Specifically, Fig. Supp. 13B shows that synaptic blockade does further increases PER, causing contradictions to authors' statements ("silencing hugin+ neurons led to enhanced sweet-driven feeding behavior (line 299-300)" and "...further silencing has little additional effect (line 402)"). The CaMPARI data (Fig. 1J) provides the link between the activity levels of hugin-releasing neurons and satiety state. However, the fact that eliminating hugin-AstA signal can promote further PER in starved flies suggests that this brake is not completely satiety-dependent. I ask authors to at least discuss this perceived discrepancy between their data and conclusions.Also, the authors' finding that PK2-R2 reduction actually suppresses PER specifically among starved flies (Fig. 3D, H), albeit with relatively small effect size, suggests that hugin-AstA axis is not a singular, linear pathway as authors suggest in Fig. Supp. 16. While delineating the PK2-R2-dependent pathway is beyond the scope of this study, at least a line of discussion would be helpful.Minor comments:(1) Fig. Supp. 8 (dTRPA1 activation of hugin and AstA neurons), and Fig. Supp. 13B-D (inhibition of hugin and AstA neurons) should be in the main figure given its relevance to the narrative of this manuscript.

We agree with the reviewer regarding their importance. The key behavioral panels from these figures have now been moved to the main figures to strengthen the narrative flow.

(2) Fig. Supp. 11 (PER and imaging using decapitated heads only), despite its creativity, leaves me wonder how PER of fly heads looks like. It is a highly artificial and invasive experiment. Supplementary movies would be helpful.

We apologize for the lack of clarity in our description. In this experiment, flies were not decapitated. Instead, we surgically severed the connection between the brain and the ventral nerve cord (VNC), while keeping the body and proboscis musculature intact. Thus, the flies remained physically intact, and PER was measured using the same behavioral protocol as in intact animals.

We have revised the figure legend to clarify this point and avoid confusion. Because the behavioral procedure was identical to standard PER assays and the flies retained normal proboscis motor function, we did not include supplementary videos.

(3) Expression patterns of PK2-R1 and AstA-R2 in proboscis are mentioned in text but with no data (lines 229 and 279). I strongly encourage authors to show images.

We have now included the relevant expression images in the revised manuscript.

(4) A citation for the "previous study (line 486)" describing PER method is required.

The appropriate citation has been added.